# Devising a Mechanism for Analyzing the Barriers of Blockchain Adoption in the Textile Supply Chain: A Sustainable Business Perspective

**Muhammad Nazam** [1] , **Muhammad Hashim** [2] , **Florian Marcel Nuță** [3] , **Liming Yao** [4] , **Muhammad Azam Zia** [5], **Muhammad Yousaf Malik** [1] , **Muhammad Usman** [6,*] and **Levente Dimen** [7,*]

1   Institute of Business Management Sciences, University of Agriculture Faisalabad, Faisalabad 38000, Pakistan
2   Faisalabad Business School, National Textile University, Faisalabad 37610, Pakistan
3   Faculty of Economics Finance and Business Administration, Universitatea Danubius Galati, 800654 Galati, Romania
4   Business School, Sichuan University, Chengdu 610064, China
5   Department of Computer Science, University of Agriculture Faisalabad, Faisalabad 38000, Pakistan
6   Institute of Agricultural and Resource Economics, University of Agriculture Faisalabad, Faisalabad 38000, Pakistan
7   Department of Cadastre, Civil Engineering and Environmental Engineering, 1 Decembrie 1918 University, 510009 Alba Iulia, Romania
*   Correspondence: usmanghani99@hotmail.com (M.U.); dimenlev@yahoo.com (L.D.)

**Abstract:** The adoption of blockchain technology (BCT) in a supply chain holds great potential for textile industries by executing transactions among stakeholders in a most reliable and verifiable way. Textile industries in emerging economies, like Pakistan, confront severe economic pressures and uncertain environment and strive to achieve sustainable supply chain excellence through blockchain implementation. This study is an initiative to analyze the key barriers in adopting BCT-related practices within the textile industry. This study conducts an extensive review of the literature using fuzzy Delphi approach for finalizing the barriers and applied fuzzy analytical hierarchy process (AHP) for prioritizing the barriers under uncertain environment. Based on the extensive review of the literature and panel discussions with experts, a total of five main barriers and 21 sub-barriers were categorized and ranked. The results and findings prioritize technological and system-related barriers (TSB) first, and human resources and R&D (HRB) barriers second among the other barrier dimensions. This paper highlights the need for an inclusive understanding of the various technological, environmental, and socio-economic perspectives to create blockchain applications that work for the textile sector. This study's key findings and policy guidelines can assist concerned stakeholders in making strategic decisions for adopting BCT within the textile supply chain. The managerial implications are provided for the industrial decision-makers and policymakers aiming to integrate BCT into the supply chain processes. Presently, there exists no research in the context of Pakistan that highlights the challenges faced during the adoption of BCT in the supply chain. For this purpose, an approach in the form of an integrated model based on fuzzy set theory is developed. Finally, the robustness of the proposed model is checked through sensitivity analysis.

**Keywords:** blockchain; barriers; supply chain; textile sector; fuzzy Delphi; fuzzy AHP

## 1. Introduction

Textile businesses are concentrating on blockchain technology as they obtain information starting from farm gate, to fibre providers, producers, distributors, wholesalers, or transporters up to retailers in a shorter time period. The organizational sustainability and compliance of goods can also be traced by taking information from the blockchain about the input supply, production phases, and distribution processes used to make them [1,2].

The textile industry is facing significant sustainability challenges in terms of sourcing methods, production and distribution of materials to its final customers. Due to technological advancements, the gap between branded items and consumers has reduced considerably. Therefore, textile and clothing organizations can easily negotiate with buyers regarding new product development even for a single garment item. The traceability feature of blockchain enables the firms to check the authenticity of any branded product for both parties, i.e., retailers and consumers. Similarly, brands can also maintain transparency with regards to their record of sales, loyalty programs and royalty payments [3,4]. This safeguards against counterfeit goods as well. Currently, industries in emerging economies are competing on price volatility and higher demand for the finest quality products. In this competitive supply chain environment, providing the finest quality items at lower prices cannot be possible without customer involvement, supplier support and integration of sophisticated technology in the organizational system [5]. Modern day supply chain networks are complex due to involvement of multi-echelon aspects and geographically disjointed entities in serving consumers [6]. Due to technological advancements in the domain of supply chain networks, it becomes practically quite difficult to evaluate business information and tackling risks in this multifaceted supply chain (SC) system [7,8].

Recently, many customers are confronted with numerous problems such as expensive products, poor quality products, and damaged or cracked products at the time of delivery due to human negligence and diverse regulatory policies in SC [9]. For these reasons, customers remain unsatisfied and provide negative feedback about services and organizations. Such disorganized business operations and disruptive issues in supply chains accordingly lead organizations toward greater trust deficits. Therefore, a need for accurate information sharing, traceability, visibility, verifiability and authenticity arises. In this way, customers lose their interest because of inferior performance and raise questions on the sustainability of SC. The verifiability and traceability are becoming emergent business challenges and a central differentiator for a number of industries in terms of managing the supply chain processes [10], such as the textile sector, automobile sector, aviation sector, pharmaceutical sector [11], and value-added goods [12]. In fact, the lack of knowledge sharing with transparency can create strategic and reputational competition issues in the supply value of any product. The higher proportionate costs incurred in tackling SC intermediaries, their reliability and transparency further complicate the traceability management of the supply chain.

The current sophisticated technologies, such as the satellite system, barcoding devices, scanning technology, radio frequency identification (RFID) tags, device components and enterprise resource planning (ERP) are mainly deployed by large organizations to keep a record of products, processes, and services across the entire supply chain, i.e., from fiber producer to brand retailer. The remaining organizations using conventional technologies and information systems have limitations in guaranteeing traceability. This brings up concerns with data manipulation and identity management of particular transactions which eventually become vulnerable to cyber-attacks. These issues are challenging when operating in uncertain environments in global supply chains. Therefore, BCT has been recognized as a potential tool that enables supply chain traceability and assists in the resolution of many related issues. The concept of BCT is described as a database that keep records of transactions, verifies the data and operates consistently between many nodes or partners in the supply chain network [13]. It may also be defined as a combination of an open distributed ledger, cryptographic algorithms and a decentralized and mechanized reconciliation system [14]. The transactions between buyer and seller through a blockchain system are recorded in a string of data blocks. These data blocks are interconnected using decentralized time-stamped algorithms in the entire supply chain [15]. Nakamoto [16] developed a novel approach for designing a blockchain payment system using an interconnected decentralized network and algorithms. Undeniably, BCT attains much popularity in the financial industrial domain, but also has an ability to create more widespread socio-economic and ecological impacts [17]. Organizational economic inefficiencies can also be minimized by

adopting BCT in bureaucratic processes. It helps minimize transaction costs, reduce human errors and reduces loss of time and other risks. This technology creates opportunities for new business dynamics and opportunities relevant to the visibility of product information and financial information in detail. Hence, it is more realistic to consider the sustainable business practices in the supply chain because data visibility and traceability have availed much intention in terms of socio-economic and ecological paradigms [18].

A significant number of problems arise in implementing BCT when considering business-to-business (B2B) perspectives. Many researchers in different disciplines and fields are focused on eradicating barriers [19–21]. They have tried but unfortunately failed to eliminate these problems due to the presence of certain factors such as centralized information systems, lack of investments on technological-oriented systems and lack of awareness of socio-economic and organizational interest in automation. Based on these particular problems, it is crucial to determine and differentiate among the barriers and risks associated with the implementation of blockchain technology to the existing businesses and their supply chain processes. Furthermore, it is necessary to suggest remedial actions for tackling or eliminating these barriers and existing risks. Additionally, these barriers influence the organizational supply chain's performance, needs, prioritization and ranking, because it is in these ways that decision makers are thought to develop their way forward in achieving business excellence. Currently, due to technological advancements, the phenomenon of integrating the BCT-related practices into supply chain processes has gained increased significance. Despite the significance of BCT, there are some constraints and difficulties encountered by textile organizations. These may relate to technological, socio-economic, human resources, R & D, organizational and individual constraints [22]. Like other technologies, blockchain considers the top-trend traceability-oriented technology that may minimize shortcomings in the supply chain network and attract the attention of researchers and practitioners [23]. There are certain types of barriers that exist; however, the key barrier is the immature technology level in the developing country context [24]. In addition, the existing inadequate technological infrastructure and systems may be considered a big challenge to the implementation of BCT [25]. Similarly, human resources and R & D are found to be a potential barrier for BCT adoption in the holistic supply chain process. The size of blockchain systems increases due to the number of transactions among partners which may shift the BCT to become more complex and highly complicated [26]. Blockchain is an expensive technology and companies may not be willing to invest in new technology due to higher risks. The supportive role of government is an essential and crucial component to promote this technology. In the developing country context, bureaucratic mindset, complex organizational set ups and hesitation of managers to disseminate transparent data may be found as key elements or variables that elongate the process of BCT based on integration of SSC. Lastly, BCT consumes excessive energy making way for environmental impacts to be included as another BCTSC-related barrier [27]. This research study is an attempt to tackle research gaps by including perspectives concerning sustainable supply chain regarding blockchain adoption barriers considering Technological and System Barriers (TSB), Human Resources and R&D Barriers (HRB) Socio-economic and Environmental Barriers (SEB), Organizational and Individual Barriers (OIB), and Governmental and External Stakeholder Barriers (GESB) in the textile sector. More specifically, this paper focuses on finding answers to several key research questions.

### 1.1. Research Questions

The present study develops the following three questions:
RQ$_1$: What are the key barriers in adoption of blockchain technology with supply chain?
RQ$_2$: What are the consequences of the identified barriers?
RQ$_3$: What are the remedial measures needed to overcome the barriers administratively?

### 1.2. Research Objectives

This study developed the below mentioned three research objectives:

$RO_1$: Identify the barriers to integrate blockchain technology into supply chain processes.

$RO_2$: Evaluate finalized barriers through different phases using the fuzzy Delphi approach based on fuzzy AHP and sensitivity analysis methods.

$RO_3$: Suggest recommendations for overcoming the barriers in terms of SCM.

In the manufacturing sector, the textile industry can be considered one of the oldest but most complex sectors as multiple types of sub-sectors exist, i.e., small and medium enterprises. The manufacturing cycle of textile-related items makes traceability features quite difficult due to the prevalence of a number of barriers in the supply chain. Therefore, the purpose of this study is to distinguish the key barriers in adopting BCT in the textile sector of Pakistan and suggest research implications for stakeholders. This work develops an integrated fuzzy model based on sustainable supply chain aspects to achieve a competitive edge in the textile market. In order to eradicate the potential barriers of BCT implementation, a multi-criteria decision-making method (MCDM) has been used. Human feelings are subjective; therefore, it is difficult to convert these feelings into numeric form in order to achieve the desired results. Therefore, to eradicate inherited subjectivity in data, this paper proposes the model based on the fuzzy set theory to solve the case of a garment-based business organization [28]. For this purpose, the fuzzy Delphi method is applied as a qualitative forecasting tool to collect the relevant data and information under the supply chain domain in a specific field [29]. The fuzzy-based AHP [30] method is used to achieve the specified goals. The judgments and assessments of the decision-makers are made according to their preference levels under vague and ambiguous environments. In addition, criteria weights and rankings are calculated by fuzzy AHP method and the validity of the proposed model is checked through sensitivity analysis. These techniques are applied to identify the intensity of importance for finalized barriers that may influence the transition to blockchain technology in supply chain processes. Given this background, this study is an attempt to understand various technological, environmental and socio-economic perspectives to align blockchain-related practices for improving business continuity of the textile sector [31]. To the best of our knowledge, there is currently a lack of research articles in the context of Pakistan that highlights the challenges and future research directions during the adoption of blockchain technology in the textile supply chain. In the existing literature, there has not been much identification of the barriers in the BCT adoption for the development of the textile industry that considers the developing country's perspective. This study also pinpoints some potential and significant obstacles for solving traceability-related issues throughout the entire supply chain, from the initial stages of procuring raw materials to the final stages of reaching consumers. For these purposes, an approach in the form of an integrated model based on fuzzy set theory has been developed in this article. The findings of this study would be applicable to blockchain applications in numerous other industries with various supply chain setups.

### 1.3. BCT-Based Research Gap in a Developing Country Context

In the emerging economies perspective, developing countries such as Pakistan are striving to increase their production capacity and investment values through automation and using value chain management based on blockchain. With the rapid changes and pressure of globalization, emerging economies are much concerned with moving forward towards productivity performance, economic efficiencies and efficient industrialization systems [32–34]. The growing marketplaces are impervious to developed nations considering that the underdeveloped countries act as catalysts in terms of simultaneously stimulating the global economy [35,36]. The increase in economic growth rates may facilitate the textile manufacturing industries of developing countries to cope with the companies of developed nations. The transparency of data related to procurement, manufacturing and transportation services in the emerging economies are becoming sophisticated and offering adequate comfort, convenience and safety measures to retail consumers [37–39]. Currently, blockchain technology has emerged as a new business paradigm that integrates global suppliers, local manufacturers and foreign customers across the world. All supply

chain partners are trying to assess the level of transparency and authenticity of business transactions in their supply chains using blockchain technology. Due to these reasons, blockchain technologies are the main pillars in terms of providing solutions to industry stakeholders. In this manner, the developing economies of nations such as Pakistan, Vietnam, Sri Lanka, Bangladesh, India and Kenya are discovering an increasing array of applications for blockchain to verify information. Blockchain is finding innovative uses in textile, pharmaceutical, banking and agriculture supply chains. Although a decade has passed since the invention of blockchain, its technology is still evolving and being tested in emerging economies.

The remaining part of this study is structured as follows: Section 2 discusses the theoretical background of BCT in the context of sustainable supply chain. Section 3 presents the proposed solution methodology based on fuzzy Delphi and fuzzy AHP approaches. In Section 4, a case is provided to demonstrate the applicability of the proposed model based on blockchain technology and integrated sustainable supply chain management. Section 5 discusses the analysis of the solutions and results. Section 6 highlights the implications of research. Finally, Section 7 concludes the paper.

## 2. Theoretical Background

This section seeks to detail the theoretical background and fundamentals of the methodologies applied in this research with the supporting literature. A detailed description of each subsection is as follows.

### 2.1. Blockchain Technology Architecture and Mechanism

The concept of blockchain can be stated as record keeping of transactions in the form of a distributed digital ledger system using encryption approaches. Basically, blockchain is a pool of decentralized transactions which maintains and verifies the data and saves the information for a certain time span [40–42]. This process is open to the accessibility level allowed by a specific company and can be accessed by its personnel at any stage to sanction the data's accuracy. It can be updated in the network through node-to-node and a new transaction can be recorded in the most efficient way. Once the system verifies the transaction, it automatically generates a block to the entire chain from upstream to downstream. After verification of the transaction, no one can change the records except the appropriate personnel [43]. In the blockchain system, consistency and accuracy among blocks is constantly monitored and updated through the distributed ledger system; therefore, it is possible to stop the threat of cyber-attacks by using this system. A fraudulent person can be easily identified if any mistake or dodge is committed. The blockchain structure can be designed through linkage between the blocks which carry the encrypted form of data, processed time and transaction status. For instance, if a new transaction needs to be processed, initially a request can be delivered to the supply chain channel managed by a peer-to-peer node in a typical supply chain network, and this request is assessed and monitored to verify the data [44]. If the forwarded request is accepted among personnel from node-to-node then it is included to the blockchain. If the request is denied, it is excluded from the blockchain and not treated as a transaction. The procedural way of data addition begins from a single stage to all the phases involved in the chain. The supply chain partners who linked the system may generate a specific code. After the generation of this code, they are eventually included in the chain [45].

In light of the above discussion, more than three services added by the blockchain system become more preferable and convenient. Firstly, the integrated blockchain system based on supply chain may minimize financial and operating costs in an efficient way. Secondly, this system accelerates operational processes through excellence without any intermediary or dependencies of workforces. Thirdly, all the processes and transactions are managed through different nodes or points which may minimize chances of fraud or losses of data leakage in the well-designed system. Finally, this system assures the transparency level of data which clarifies the flow of products from the point of origin to

the end source by indicating the visibility of the product, people and places. Hence, based on this argument, blockchain technology is considered an integral part of a value chain, new product development and technological advancement with regards to solving major difficulties encountered by stakeholders and major beneficiaries. In Figure 1, the typical structure and mechanism of blockchain showing the numbers of nodes can accept or reject modifications in the entire supply chain process.

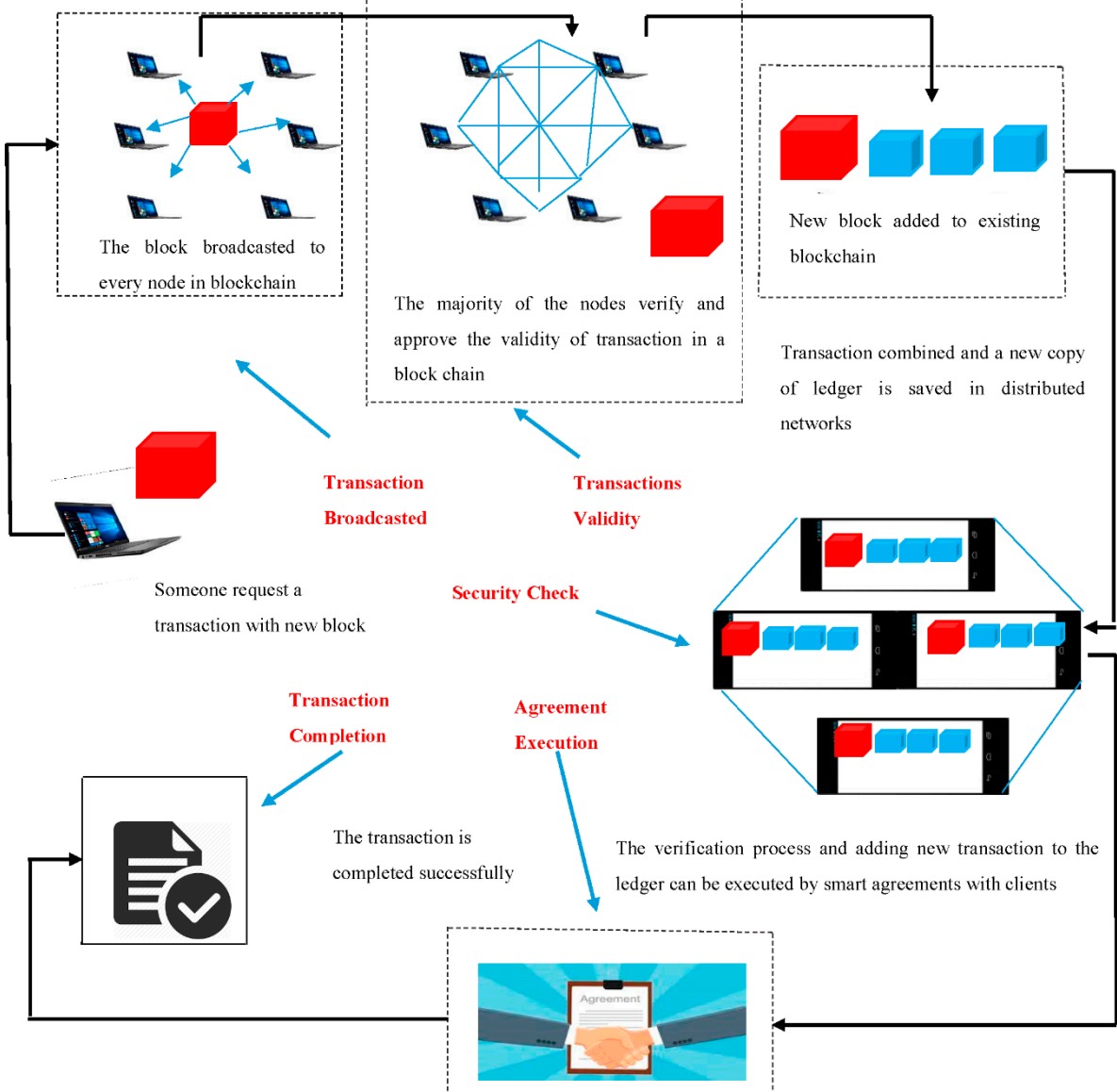

**Figure 1.** Blockchain architecture and functioning mechanism in a typical supply chain.

## 2.2. Blockchain Technology Integrated with Sustainable Supply Chain (BCTSC)

Blockchain technology can generate database management systems for data collection, data saving, data analysis and significant information regarding traceability, neutrality and reliability in the technological context [46]. Chang and Chen [47] conducted a detailed systematic review of the literature on the latest developments regarding blockchain practices and applications. They suggested that blockchain practices can be a helping tool in meeting the supply chain objectives in a sustainable way. Similar research was also conducted by Lim et al. [48] on a review of the literature of BCT applications in supply chains, and they presented comprehensive analysis using different themes, methodologies and industries. Blockchain technology integrated with sustainable supply chains (BCTSC)

and internet of things (loT) have caught the attention of industrial stakeholders, academia and policy makers [15,49,50]. It does not merely focus on economic aspects in supply chain management but also on the targeting of environmental and social domains in a more generalized way. The implementation of blockchain technology would be a remedial measure for tackling high levels of complexity in sustainability such as financial, communal and ecological aspects. Therefore, capturing the triple bottom line might exemplify the extensiveness of blockchain technology application as the traceability of socio-economic and ecological trends may become the root cause for occupational health and safety concerns in the blockchain [51]. Paliwal et al. [12] developed a framework for implementing BCT in managing the supply chain and suggested the way for achieving sustainability. Erol et al. [52] assessed the feasibility of blockchain technology in different industries of Turkey. Siegfried et al. [53] integrated BCT with industrial internet of things for improving the trackability using systematic fit analysis. Due to the transparency feature of blockchain, human rights and best work practices can be assured by the companies. Similarly, throughout the entire supply chain, the product flow assures the authenticity of a product to the customers. This transparency across the supply chain, from upstream to downstream, is considered an ethically strong point.

### 2.2.1. Economic Perspective

The ready-made garments and textile organizations encounter sustainability related issues in the supply chain. For achieving sustainable supply chain excellence, the textile industry needs to link with blockchain technology such as radio frequency identification (RFID) to ensure the traceability process. In blockchain technology, information can be maintained by persons who may enhance the chances of detecting unethical suppliers and the supply of counterfeit products in the market [54]. In addition, blockchain technology can reduce the chances for human errors, the cost of preventing data, supply chain failures and transaction times. The clothing and textile-based multinational companies realized the essence of transparency and visibility as a competitive edge to win the trust of customers and potentially increase their revenue [39].

### 2.2.2. Social Perspective

Blockchain technology also aids socially and contributes to the supply chain by making the available information more stable among the participants [55]. The recorded information cannot be fluctuated or changed, which may enhance the chance to identify corrupt officials by enabling regulatory organizations or bodies to detect the culprits who behave unfairly, socially or morally speaking. This technology can stop the frequency of cyberattacks and block the notorious intermediary agents who mislead companies [56]. An interesting feature of blockchain is traceability which facilitates the process of formulating the strategies for better human rights, best work practices and fair reward systems. The transparent flow of information regarding product history builds the confidence level of customers in the entire supply chain [57].

### 2.2.3. Environmental Perspective

According to environmental perspectives, blockchain technology contributes to reduce the rework and recall practices within the supply chain [58,59]. Blockchain technology can reduce energy storage, consumption and emissions which conserve natural resources. Sternberg et al. [60] conducted an empirical study and derived some significant insights from the adoption of blockchain in a typical supply chain. Bai and Sarkis (2020) [61] developed a supply chain transparency appraisal model by considering the sustainable aspects. The traditional systems lack the transparency element to deliver eco-friendly products due to a lack of information and distribution system for green products. Once the production and distribution systems of green products are verified and confirmed, customers can show their interest in buying products on a priority basis [62]. Blockchain technology integrated with sustainable supply chain management may prove advantageous

for an emissions trading program (ETP) by enriching the effectiveness of emission trading schemes (ETS). Therefore, the stakeholders in a supply chain pay attention to the positive aspects of blockchains rather than the negative aspects, as doing so may influence the significance of supply chain sustainability.

### 2.3. Understanding BCTSC in the Textile Industry and Potential Barriers

Blockchain technology reshaped business patterns from farming practices to finished garments across the world. A few years back, the textile industry was not in a position to transform accurate information to concerned stakeholders in the entire supply chain ranging from farmers to end consumers. The significant feature of BCT is decentralization, which can reduce the probability of failure among supply chain partners as it is not dependent on an individual business entity. The blockchain technology in the textile sector works on the principle of generating an irreplaceable physical-digital link between goods and their digital identities during business transactions. Normally, a serial number is used as a physical identifier in order to link back to the single product's "digital twin." The generated link shares information transparently among supply chain partners in the textile sector [63]. For instance, if physical-digital links are not found, it means the transaction relates to counterfeit goods and the products are not genuine. In this scenario, due to a lack of transparent information or missing digital links, authentic products were diverted and replaced with counterfeit goods that can be carried through the entire value chain system unidentified. In the textile industry, blockchain technology offers significant benefits in terms of fair policies, reward systems, product flow systems, compliance, transparency, error identification and payment procedures. Nandi et al. [64] redesigned supply chains using blockchain technology with circular economy perspectives and also considered the impact of COVID-19 on supply chain sustainability. Despite the potential benefits of blockchain, there are few companies that have implemented BCT in the textiles supply chain context. The manufacturers of textile products include various processes, systems and transactions from field to fabric. BCTSC is an integral part of sustainable businesses in today's competitive global marketplace. In recent years, textile exporters in Pakistan were not conscious of BCTSC due to the lack of knowledge on BCTSC and fear of financial burden due to implementing an entire supply chain. Presently, textile exporters are concerned regarding the integration of BCTSC activities and fundamental difficulties faced by organizations within the textile sector. This research study discusses the barriers linked with the implementation of BCT. In an uncertain environment, imbalances and instabilities may arise within an SC system due to the existence of unveiled barriers, i.e., logistics contracts, sustainability issues, child labor problems and ensuring carbon emission standards. In this context, blockchain technology may be used as a key tool for tackling all identified major issues. Information visibility may authorize customers with regards to socio-economic and environmental perspectives. A well-integrated system based on BCT processes can create a more efficient and effective supply chain network system. Niranjanamurthy et al. [65] conducted a systematic literature review and analysis for understanding BCT prospects and challenges while considering future aspects. Despite the benefits of this technology, there are some technological, economic, human resource based and industrial shortcomings and barriers in the combination of blockchain with sustainable supply chain. There is a need to address these barriers to establish a significant level of BCT integrated with supply chain management. Therefore, these barriers can be categorized and evaluated under five major domains: technological and system barriers (TSB), human resources and R&D barriers (HRB), socio-economic and environmental barriers (SEB), organizational and individual barriers (OIB) and governmental and external stakeholder barriers (GESB).

### 2.4. Problem Statement and Research Highlights

In Pakistan, the textile industry experiences a lot of factors that affect the productivity and order fulfillment for customers. The major reasons for these issues are related to

little value addition, unskilled or semi-skilled labor, quality management, technological advancements and lack of modern machinery, etc. In the textile supply chain, the ready-made garments sector is a growing sub-sector that plays a significant function in the economic growth of the country. This sector employs a major portion of the workforce and contributes about 67 percent to the export business of Pakistan. The textile and apparel sector plays a vital role in Pakistan's economy, and is one of the largest sectors in Pakistan and contributes 46% to total manufacturing. This sector contributes almost 9.5% of the gross domestic product (GDP) of Pakistan and engages around 45% of the total workforce in the country. The textile sector employs 38% of the labor force of the country and contributes 67% in exports. These facts and figures depict Pakistan as the fourth major manufacturer and exporter of cotton products and having the third largest spinning capacity in the Asia-Pacific region after China, Bangladesh and India and also contributes 5% to the worldwide capacity of the spinning sector.

In this context, this research study addresses key barriers faced by the textile industry in adoption of blockchain technology integrated with the sustainable supply chain system. Few textile industries are trying to adopt this technology but most of the industries are unable to implement the technology due to a certain number of barriers. The literature on BCTSC is not sufficient and only a few studies are available in the context of the textile industry of Pakistan. In this sense, integrating blockchain technology with supply chain networks can be beneficial for increasing revenue and improving the flow of information among fiber providers, manufacturers, processors, shippers and retailers in no time. The integration of BCTSC may also include several barriers in the textile industry; therefore, these barriers need to be explored and evaluated to confirm the efficiency and effectiveness of resources. The empirical studies conducted on the identification of barriers in adopting (BCT) are to some extent unfledged in the textile sector, as compared to other sectors such as pharmaceuticals, electronics, and agriculture etc.

The salient contributions of this research study are as follows:

- This study bridged the research gap in the BCTSC agenda by identifying the key barriers in implementation of BCT in the textile sector along with an extensive review of the literature and feedback from experts using fuzzy Delphi technique.
- The identified barriers were analyzed using fuzzy AHP for acquiring priority weight vector which would facilitate decision makers in grasping their relative significance in terms of understanding BCTSC in the textile industry.

The research discussed managerial and social implications in detail which are based on the study's results and findings. In the context of the Asia-Pacific region, it is a novel approach and addressing barriers on BCTSC would facilitate the industries to improve the sustainable business performance not only in domestic markets but also globally. This study develops a skeletal model of BCTSC activities that would help policymakers and concerned stakeholders to adopt BCT activities in their businesses. The categorization of barriers relevant to existing works based on relevant studies are provided in Table 1.

*2.5. Motivation of the Study*

After conducting an extensive review of the contemporary literature related to BCT, it was revealed that most of the research studies overlooked comprehensive findings and adoptability measures to overcome barriers for the successful adoption of BCT in the textile industry of Pakistan. The few existing studies that discuss the use of cryptocurrencies have not received much attention by concerned stakeholders in order to explore the existing barriers in textile firms. In the developing country context, few such studies are available that investigate the detailed categorization of BCT-related barriers in the entire ecosystem of the textile supply chain. In this sense, integrating blockchain technology with supply chain networks can be beneficial for increasing revenue and improving the flow of information among fiber providers, manufacturers, processors, shippers and retailers in no time. In real life scenarios and practical cases, it has been experienced that it is not an easy task to concentrate on all types of barriers instantaneously, because this type of study bears much

financial budgeting, technical staff, and time constraints etc. The review of the comprehensive literature and the integrated BCTSC model indicate the relevance of a total of five major dimensions/contexts with 21 barriers to influence blockchain technology adoption in textile supply chains. Based on previous studies conducted by various researchers, it has been supposed that the results of the present study would be beneficial to a larger audience and professionals who are involved in working on BCT platforms [87]. Therefore, the motivation behind conducting this present study is that we firstly assess the most critical ones by using fuzzy Delphi approach to address the uncertain factors while finalizing the barriers through consultation with a team of experts and then prioritizing in descending order category-wise.

**Table 1.** Categorization of key barriers to adopt BCT in a sustainable supply chain.

| Categories of Barriers | Codes | Key Barriers to Implement BCT in SSC | Source |
|---|---|---|---|
| Technological and System Barriers (TSB) | $TSB_1$ | Immaturity of technology | [66] |
| | $TSB_2$ | Transparency and traceability-related issues | [67] |
| | $TSB_3$ | Challenges in sustainable practices and blockchain technology | [68] |
| | $TSB_4$ | Risks of cyber attacks | [69] |
| | $TSB_5$ | Lack of expertise and technical support in IT | [70] |
| Human Resources and R&D Barriers (HRB) | $HRB_1$ | Negative perception of IT among workers' minds | [71] |
| | $HRB_2$ | Lack of professional technical labor | [72] |
| | $HRB_3$ | Lack of research and development departments | [73] |
| | $HRB_4$ | Lack of financial resources for technological infrastructure | [74] |
| Socio-economic and Environmental Barriers (SEB) | $SEB_1$ | Poor economic behavior in the long run | [75] |
| | $SEB_2$ | Social and cultural constraints | [76] |
| | $SEB_3$ | Neglecting environmental standards | [77] |
| Organizational and Individual Barriers (OIB) | $OIB_1$ | Lack of management commitment and support | [78] |
| | $OIB_2$ | Information sharing obstacles | [79] |
| | $OIB_3$ | Lack of organizational new policies for adopting technology | [80] |
| | $OIB_4$ | Employees reluctance to accept change | [81] |
| | $OIB_5$ | Unwillingness to change the conventional ways | [82] |
| Governmental and External Stakeholder Barriers (GESB) | $GESB_1$ | Unclear issue of taxation and regulatory uncertainties | [83] |
| | $GESB_2$ | Lack of government interest in blockchain | [84] |
| | $GESB_3$ | Market competition and uncertainty | [85] |
| | $GESB_4$ | Lack of external stakeholders' involvement | [86] |

## 3. Solution Methodology

### 3.1. Research Design

The process for screening the review of literature was taken by considering the objectives of research that examined the previous studies on blockchain applications in the context of the textile supply chain (TSC). For this purpose, multiple databases were searched, including IEEE, Scopus and Web of Science, for the terms such as blockchain, textile supply chain and sustainability, as well as their different dimensions, which include traceability, transparency, information sharing, technological infrastructure, holistic supply chain, network capabilities, innovation and data access. The search resulted in 59 research studies describing the paradigms of blockchain, sustainability and the TSC. Following a thorough examination of these articles, 41 barriers to blockchain adoption in the TSC were identified as were research gaps in the context of emerging economies. Furthermore, the listed barriers on the basis of the literature review are discussed with expert teams for finalizing the key barriers in the context of the textile industry in Pakistan. A total number of ten experts, i.e., from the industry and academia participated to ensure proper assessment of these barriers. The experts, i.e., decision makers from academia and the textile

industry were approached to assess the intensity of the barriers through a well-structured questionnaire using a linguistic scale as demonstrated in Table 2. The group of decision makers was finalized using a convenience sampling technique. In the initial phase, a fuzzy Delphi approach was utilized to finalize the barriers in BCT adoption. In the preliminary phase, a team of ten experts comprising a senior IT manager, a design manager, a director of operations, a supply chain manager, a project manager, a general manager of production, a supply chain professional, a chief information officer and two professionals from academia, (i.e., a professor and an associate professor) were assigned to confirm the suitability level of identified barriers to adopt BCT in the textile industry. The industrial experts were enriched with more than 10 years of experience and have expertise in the field of supply chains and BCT. The academic experts have sufficient academic subject knowledge and research experience. The fuzzy Delphi approach was utilized to defuzzify the experts' inputs into crisp values. The results for finalizing barriers in blockchain integrated with supply chain are provided in Table 3. The fuzzy pairwise comparison matrix of main barriers is provided in Table 4. Based on experts' input and the results of the fuzzy Delphi method, 21 barriers were listed in this research study and were further classified into five major barrier dimensions. In the second phase, after finalizing the group of experts and barrier dimensions, fuzzy AHP method was applied to construct a pairwise comparison matrix. A well-structured questionnaire was circulated among experts to seek information for the formulation of pairwise comparison matrices for the main barriers and sub-barrier categories using Saaty's scale. The data were collected from those experts who have diverse industrial backgrounds and knowledge of supply chains. This process was complicated as only a few experts were available with sufficient knowledge and information, and with real-life experience in the implementation of blockchain in the TSC. We set our criteria in such a way that experts should possess understanding of blockchain and should have had the experience of adopting the technology in the TSC. The decision makers must be neutral by not preferring a particular blockchain barrier. For accomplishing the objectives of this study, a total of ten experts participated in the data collection process. Previous research studies used sample sizes of three, five, and nine experts. The authenticity in the selection of experts is in accordance with previous and similar research. Moreover, based on the expertise and designation of experts, it was requested that they provide input regarding the importance level of barriers ranging from "very low" to "very high" as values provided in Table 2. The pairwise comparison matrix was constructed for the barrier categories based on the judgment of an expert panel. The final barriers, weights and ranking of barrier dimension were obtained as shown in Tables 5 and 6.

**Table 2.** Linguistic scale applied in this study.

| Linguistic Variables | Notations | Fuzzy Number | Fuzzy Score |
|---|---|---|---|
| Very Low | VL | $\tilde{1}$ | (0, 0, 0.1) |
| Low | L | $\tilde{2}$ | (0, 0.1, 0.3) |
| Medium Low | ML | $\tilde{3}$ | (0.1, 0.3, 0.5) |
| Medium | M | $\tilde{4}$ | (0.3, 0.5, 0.7) |
| Medium High | MH | $\tilde{5}$ | (0.5, 0.7, 0.9) |
| High | H | $\tilde{6}$ | (0.7, 0.9, 1.0) |
| Very High | VH | $\tilde{7}$ | (0.9, 1.0, 1.0) |

The proposed problem is divided into three main hierarchical phases and schematically outlined in Figure 2. The identified phases are (Phase-1) the application of fuzzy Delphi approach to determine the barriers categories, (Phase-2) the computation of the criterion weight using fuzzy AHP methodology, and (Phase-3) the checking of fluctuations among results and suggesting remedial measures using sensitivity analysis for adoption of BCT in SSC. Figure 2 also provides procedural guidelines to industrial decision makers and policymakers aiming to integrate BCT into supply chain processes.

**Table 3.** Fuzzy Delphi method analysis for finalizing barriers in blockchain integrated with supply chain.

| Barriers | Fuzzy Weight | Defuzzification | S/R |
|---|---|---|---|
| 1 | (0.30, 0.82, 1.00) | 0.7067 | S |
| 2 | (0.00, 0.48, 1.00) | 0.4933 | R |
| 3 | (0.00, 0.40, 1.00) | 0.4667 | R |
| 4 | (0.30, 0.84, 1.00) | 0.7133 | S |
| 5 | (0.10, 0.70, 1.00) | 0.6000 | S |
| 6 | (0.10, 0.58, 1.00) | 0.5600 | R |
| 7 | (0.00, 0.44, 1.00) | 0.4800 | R |
| 8 | (0.00, 0.80, 1.00) | 0.6000 | S |
| 9 | (0.10, 0.84, 1.00) | 0.6467 | S |
| 10 | (0.30, 0.74, 1.00) | 0.6800 | S |
| 11 | (0.00, 0.52, 1.00) | 0.5067 | R |
| 12 | (0.30, 0.90, 1.00) | 0.7333 | S |
| 13 | (0.00, 0.58, 1.00) | 0.5267 | R |
| 14 | (0.30, 0.72, 1.00) | 0.6733 | S |
| 15 | (0.00, 0.58, 1.00) | 0.5267 | R |
| 16 | (0.10, 0.76, 1.00) | 0.6200 | S |
| 17 | (0.70, 0.94, 1.00) | 0.8800 | S |
| 18 | (0.00, 0.42, 1.00) | 0.4733 | R |
| 19 | (0.00, 0.30, 0.70) | 0.3333 | R |
| 20 | (0.10, 0.84, 1.00) | 0.6467 | S |
| 21 | (0.30, 0.82, 1.00) | 0.7067 | S |
| 22 | (0.00, 0.36, 1.00) | 0.4533 | R |
| 23 | (0.50, 0.90, 1.00) | 0.8000 | S |
| 24 | (0.00, 0.50, 1.00) | 0.5000 | R |
| 25 | (0.10, 0.78, 1.00) | 0.6267 | S |
| 26 | (0.00, 0.46, 1.00) | 0.4867 | R |
| 27 | (0.10, 0.64, 1.00) | 0.5800 | R |
| 28 | (0.30, 0.86, 1.00) | 0.7200 | S |
| 29 | (0.00, 0.80, 1.00) | 0.6000 | S |
| 30 | (0.00, 0.22, 0.70) | 0.3067 | R |
| 31 | (0.00, 0.10, 0.50) | 0.2000 | R |
| 32 | (0.00, 0.34, 0.90) | 0.4133 | R |
| 33 | (0.10, 0.84, 1.00) | 0.6467 | S |
| 34 | (0.00, 0.80, 1.00) | 0.6000 | S |
| 35 | (0.10, 0.76, 1.00) | 0.6200 | S |
| 36 | (0.00, 0.24, 0.50) | 0.2467 | R |
| 37 | (0.00, 0.50, 0.90) | 0.4667 | R |
| 38 | (0.50, 0.88, 1.00) | 0.7933 | S |
| 39 | (0.70, 0.96, 1.00) | 0.8867 | S |
| 40 | (0.00, 0.34, 0.70) | 0.3467 | R |
| 41 | (0.00, 0.60, 1.00) | 0.5333 | R |

Note: S—Selected and R—Rejected.

**Table 4.** Fuzzy pairwise comparison matrix of main barriers.

| | TSB | HRB | SEB | OIB | GSEB |
|---|---|---|---|---|---|
| TSB | (1.00, 1.00, 1.00) | (3.00, 5.40, 9.00) | (0.20, 2.04, 3.00) | (0.14, 2.29, 5.00) | (0.14, 1.57, 7.00) |
| HRB | (0.11, 0.19, 0.33) | (1.00, 1.00, 1.00) | (0.14, 2.82, 5.00) | (0.20, 3.64, 7.00) | (3.00, 3.80, 7.00) |
| SEB | (0.33, 0.49, 5.00) | (0.20, 0.35, 7.00) | (1.00, 1.00, 1.00) | (0.20, 3.64, 7.00) | (0.14, 2.69, 5.00) |
| OIB | (0.20, 0.44, 7.00) | (0.14, 0.27, 5.00) | (0.14, 0.27, 5.00) | (1.00, 1.00, 1.00) | (0.14, 2.26, 5.00) |
| GSEB | (0.14, 0.64, 7.00) | (0.14, 0.26, 0.33) | (0.20, 0.44, 7.00) | (0.20, 0.44, 7.00) | (1.00, 1.00, 1.00) |

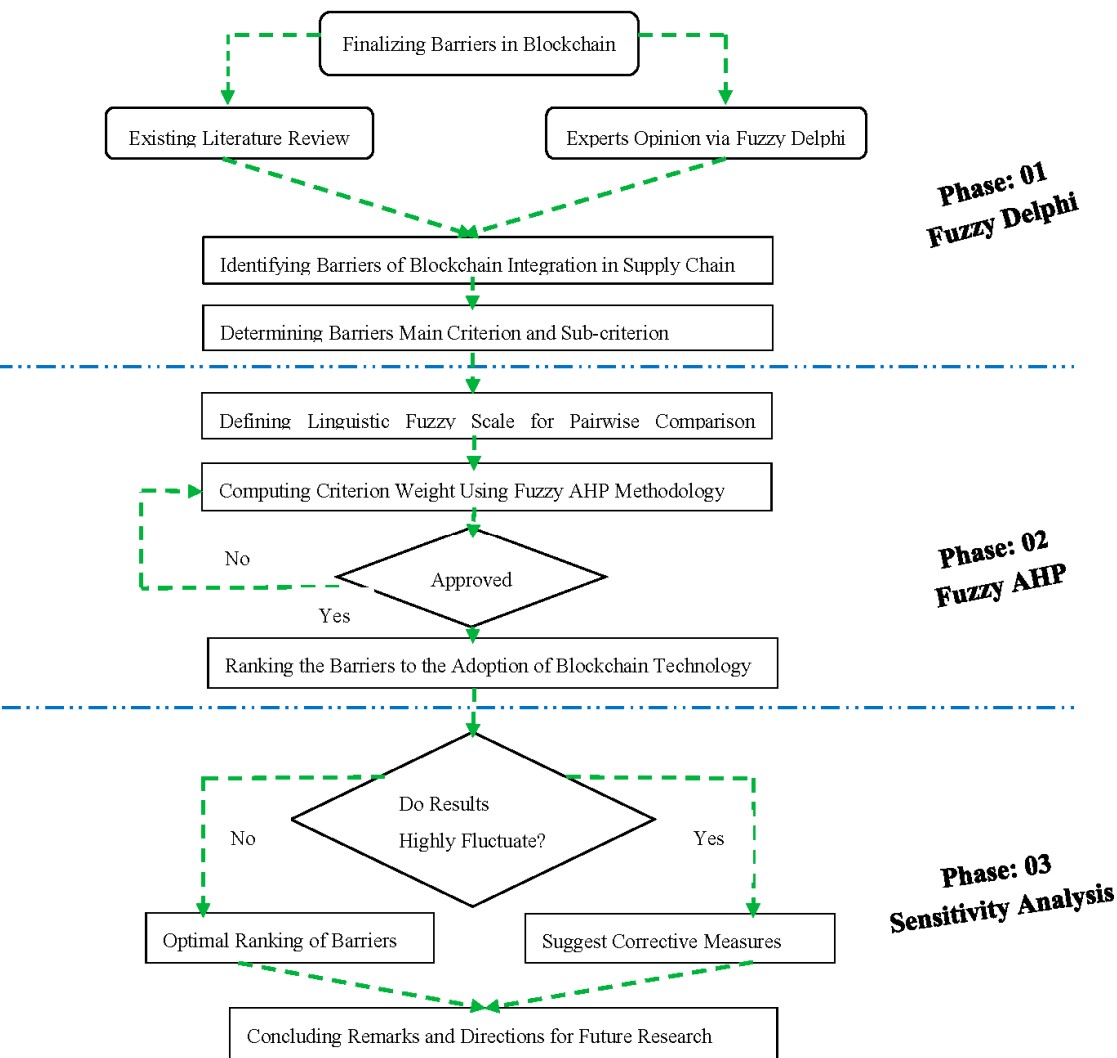

**Figure 2.** Proposed Research Framework to Integrate the Blockchain with Sustainable Supply Chain.

**Table 5.** Ranking of dimension of barriers.

|  | Weight | Ranking |
|---|---|---|
| TSB | 0.233 | 1 |
| HRB | 0.220 | 2 |
| SEB | 0.194 | 3 |
| OIB | 0.179 | 4 |
| GSEB | 0.173 | 5 |

### 3.2. Fuzzy Set Theory

Most of the time, our need to make a decision encounters a scenario where sufficient information is not available. We normally face environmental uncertainty, and it becomes difficult to reach a certain decision. In such a situation, the fuzzy set theory supports managers [88]. In this study, we used triangular fuzzy number which is widely used by scholars [89,90]. The membership function is discussed in Equation (1):

$$\mu_C(\chi) = \begin{Bmatrix} 0, & x \leq p \\ \frac{x-p}{q-p}, & x \in [p,q] \\ \frac{x-r}{q-r}, & x \in [q,r] \\ 0, & Otherwise \end{Bmatrix} \tag{1}$$

**Table 6.** Final ranking of barriers.

| Barrier Category | Main Barriers Weight | Sub-Barriers | Sub-Barriers Weight | Final Weight | Ranking |
|---|---|---|---|---|---|
| TSB | 0.233 | $TSB_1$ | 0.4511 | 0.1052 | 3 |
| | | $TSB_2$ | 0.2901 | 0.0676 | 6 |
| | | $TSB_3$ | 0.1422 | 0.0332 | 11 |
| | | $TSB_4$ | 0.0748 | 0.0175 | 16 |
| | | $TSB_5$ | 0.0417 | 0.0097 | 20 |
| HRB | 0.220 | $HRB_1$ | 0.5545 | 0.1222 | 2 |
| | | $HRB_2$ | 0.2474 | 0.0545 | 7 |
| | | $HRB_3$ | 0.1257 | 0.0277 | 12 |
| | | $HRB_4$ | 0.0724 | 0.0159 | 17 |
| SEB | 0.194 | $SEB_1$ | 0.6378 | 0.1237 | 1 |
| | | $SEB_2$ | 0.2577 | 0.0500 | 8 |
| | | $SEB_3$ | 0.1045 | 0.0203 | 15 |
| OIB | 0.179 | $OIB_1$ | 0.4683 | 0.0838 | 5 |
| | | $OIB_2$ | 0.2542 | 0.0455 | 9 |
| | | $OIB_3$ | 0.0500 | 0.0089 | 21 |
| | | $OIB_4$ | 0.1476 | 0.0264 | 13 |
| | | $OIB_5$ | 0.0800 | 0.0143 | 18 |
| GSEB | 0.173 | $GSEB_1$ | 0.5812 | 0.1008 | 4 |
| | | $GSEB_2$ | 0.0638 | 0.0111 | 19 |
| | | $GSEB_3$ | 0.1211 | 0.0210 | 14 |
| | | $GSEB_4$ | 0.2383 | 0.0413 | 10 |

*3.3. Fuzzy Delphi Method*

Fuzzy Delphi Method is a research tool that is useful for collecting information or experts' views on certain issues. The fuzzy Delphi method was first developed by Ishikawa [91] in 1993. It is very useful for capturing the uncertainty in data and has been used in multiple areas such as assessment of organizational productivity, adoption of technology and development of vendor and operations management. Based on the previous literature, fuzzy Delphi technique has been used to solve problems of judgement, with the objective of analyzing the barriers in the sustainable supply chain of the textile sector in Pakistan. The detailed process is discussed below:

Step 1: In the first step, the categorization of barriers related to the BCT-based study in the supply chain is employed. In this problem, the previous and current literature-based barriers to BCT implementation in the textile sector were identified.

Step 2: This step deals with the identified barriers and focuses on a collection of expert input. The experts, i.e., decision makers from academia and the textile industry were approached to assess the intensity of barriers through a well-structured questionnaire using a linguistic scale value as demonstrated in Table 2. The decision makers group was finalized using a geometric mean model. In this study, triangular fuzzy numbers were used for the assessment of barriers. Let us assume that fuzzy number $\widetilde{z}_{ij}$ to be the jth barrier analysis of the ith expert of k number of experts.

$$\widetilde{z}_{ij} = \left(a_{ij}, b_{ij}, c_{ij}\right) \text{ for } i = 1, 2, 3, \ldots, n \text{ and } j = 1, 2, 3, \ldots, m$$

Afterwards, the fuzzy weight vector of barriers $\widetilde{a}_j$ are given as follows: $\widetilde{a}_j = \left(a_j, b_j, c_j\right)$, where

$$a_j = \min(a_{ij}),$$

$$b_j = \left(\prod_{i=1}^{n} (b_{ij})\right)^{1/n}$$

$$c_j = \max(c_{ij})$$

Step 3: The last step in Delphi method is relevant to the determination of crucial barriers by comparing each barrier's weight with the threshold value of ($\alpha$). Furthermore, the fuzzy average value of each barrier is defuzzified into a crisp value ($S_j$) by applying the below equation.

$$S_j = (a_j + b_j + c_j)/3,$$

The acceptance or rejection of barriers based on an adjusted threshold value of ($\alpha$) set. (1) If $S_j \geq \alpha$ accepts the barriers for further analysis; (2) If $S_j < \alpha$ rejects the acceptance of the barriers and excludes them from the list of barriers.

### 3.4. Fuzzy Analytic Hierarchy Process (FAHP)

FAHP is a very useful tool for ranking or categorizing the factors being studied on the basis of their fuzzy rating scales. In this method, we request experts to provide their assessment in the form of linguistic fuzzy numbers which show the significance level of one barrier type as a criterion over the other, and the optimal alternative is selected by applying algebraic operators. The detailed process of fuzzy AHP method is discussed as follows:

Step 1: Industrial and academic experts are contacted for their responses on our constructed questionnaire on Saaty's scale.

Step 2: The responses of all experts are integrated on the basis of geometric mean method for making the matrix of pair-wise assessment. The input values of experts in terms of analyzing the ratings of criterion are provided as follows (see Equation (2)).

$$\left(\widetilde{x}_{ij}\right) = \left(a_{ij}, b_{ij}, c_{ij}\right)$$

$$a_{ij} = \underset{k}{min}\left(a_{ijk}\right), b_{ij} = \frac{1}{K} * \sum_{k=1}^{K}(a_{ijk}), C_{ij} = \underset{k}{man}\left(a_{ijk}\right) \tag{2}$$

where $i = 1, 2, \dots, n, j = 1, 2, \dots, m$, and $k = 1, 2, \dots, K$ number of experts.

Step 3: One of the most popular methods developed known as Chang's Extent Analysis technique is employed for transforming fuzzy numbers into crisp values and providing the priority weights of barriers. In this research study, a fuzzy extent analysis value is calculated for each of the criterion with respect to the goal assigned in the decision hierarchy by considering triangular fuzzy numbers (TFNs). The value of m indicates the number of extent analysis values for each criterion which is depicted as follows:

$$M_{g_t}^1, M_{g_t}^2, \dots, M_{(g_t)}^m, i = 1, 2, \dots, n$$

where, $M_{g_t}^j (j = 1, 2, \dots, m)$ are all TFNs

Next step is to compute the fuzzy extent value w.r.t. the $i$th object.

$$S_i = \sum_{j=1}^{m} M_{g_t}^j \otimes \left[S_i = \sum_{i=1}^{n}\sum_{j=1}^{m} M_{g^t}^j\right]^{-1} \tag{3}$$

In order to obtain $\sum_{j=1}^{m} M_{g_t}^j$, the fuzzy additive operations need to be performed as given below:

$$\sum_{j=1}^{m} M_{g_t}^j \left(\sum_{j=1}^{m} p_j, \sum_{j=1}^{m} q_j, \sum_{j=1}^{m} r_j\right) \tag{4}$$

For obtaining

$$\left[\sum_{i=1}^{n}\sum_{j=1}^{m} M_{g^t}^j\right]^{-1}$$

It is essential to compute the fuzzy addition operation with $M_{g^t}^j$ ($j = 1, 2, \ldots, m$) values.

$$\sum_{i=1}^{n}\sum_{j=1}^{m} M_{g^t}^j = \left(\sum_{j=1}^{m} p_j, \sum_{j=1}^{m} q_j, \sum_{j=1}^{m} r_j\right) \quad (5)$$

Then, the inverse of the identified vector is calculated by substituting the values in Equation (3), so that

$$\left[\sum_{i=1}^{n}\sum_{j=1}^{m} M_{g^t}^j\right]^{-1} = \left[1/\sum_{i=1}^{n} r_i, 1/\sum_{i=1}^{n} q_i, 1/\sum_{j=1}^{n} p_j\right] \quad (6)$$

The degree of possibilities of $M_2 = (p_2, q_2, r_2) \geq M_1 = (p_1, q_1, r_1)$ is defined as

$$V(M_2 \geq M_1) = \sup_{y \geq x}\left[\min\left(\mu_{M_1}(x), \mu_{M_2}(y)\right)\right] \quad (7)$$

This can be expressed as follows:

$$V(M_2 \geq M_1) = hgt(M_1 \cap M_2) = \mu_{M_2}(d)$$

$$= \begin{cases} 1 & if\ q_2 \geq q_1 \\ 0 & if\ q_2 \geq q_1 \\ p_1 - r_2/(q_2 - r_2) - (q_1 - p_1) & Otherwise \end{cases} \quad (8)$$

Figure 3 demonstrates the intersection between two TFNs where $d$ is the ordinate value of the highest intersection point D between $\mu M_1$ and $\mu M_2$. Therefore, using the ordinate value characteristics of fuzzy set; $M_i$ ($i = 1, 2, \ldots, k$) is stated by

$$V(M \geq M_1, M_2, \ldots, M_k) = V[(M \geq M_1)\ and\ (M \geq M_2)\ and$$

$$(M \geq M_k)] = minV(M \geq M_i),\ i = 1, 2, 3, \ldots, k \quad (9)$$

*Assume that $d'(A_i) = min\ V(S_i \geq S_k)$ for $k = 1, 2, \ldots, n$; $k = i$ and weights are*

where $A_i$ ($i = 1, 2, \ldots, n$) are $n$ elements.

$$W' = \left(d'(A_1), d'(A_2), \ldots, d'(A_n)\right) \quad (10)$$

Step 4: Normalizations of weights

$$W = (d(A_1), d(A_2), \ldots, d(A_n))^T \quad (11)$$

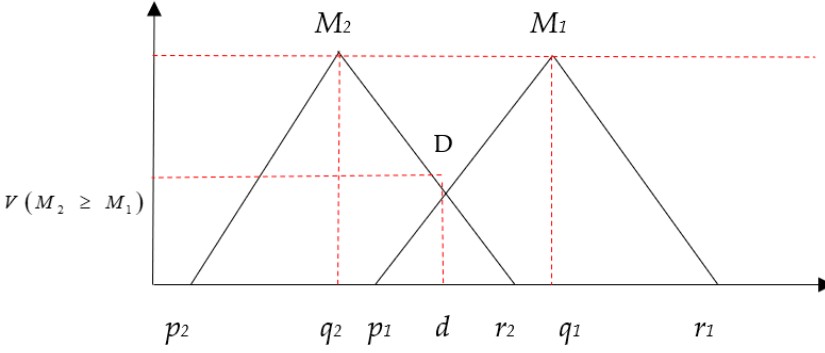

**Figure 3.** Intersection between $M_2$ and $M_1$.

## 4. An Illustrative Case

The present research is targeted on a case of textile supply chain to address the barriers encountered in adopting blockchain technology. Additionally, the case of textile supply chain is taken as a part of *Sustainable Supply Chain Excellence of Garment Division* as a strategic policy to implement BCT practices to enhance its business continuity for a longer time period in the uncertain supply chain environment. Therefore, a team of 10 experts from academia and the textile industry was formulated to seek responses on the proposed problems. The experts had expertise in the discipline of operations and logistics as well as information technology management. The hierarchical structure for the prioritization of barriers in blockchain integrated supply chain is shown in Figure 4. The procedural phases of developed methodology is discussed below:

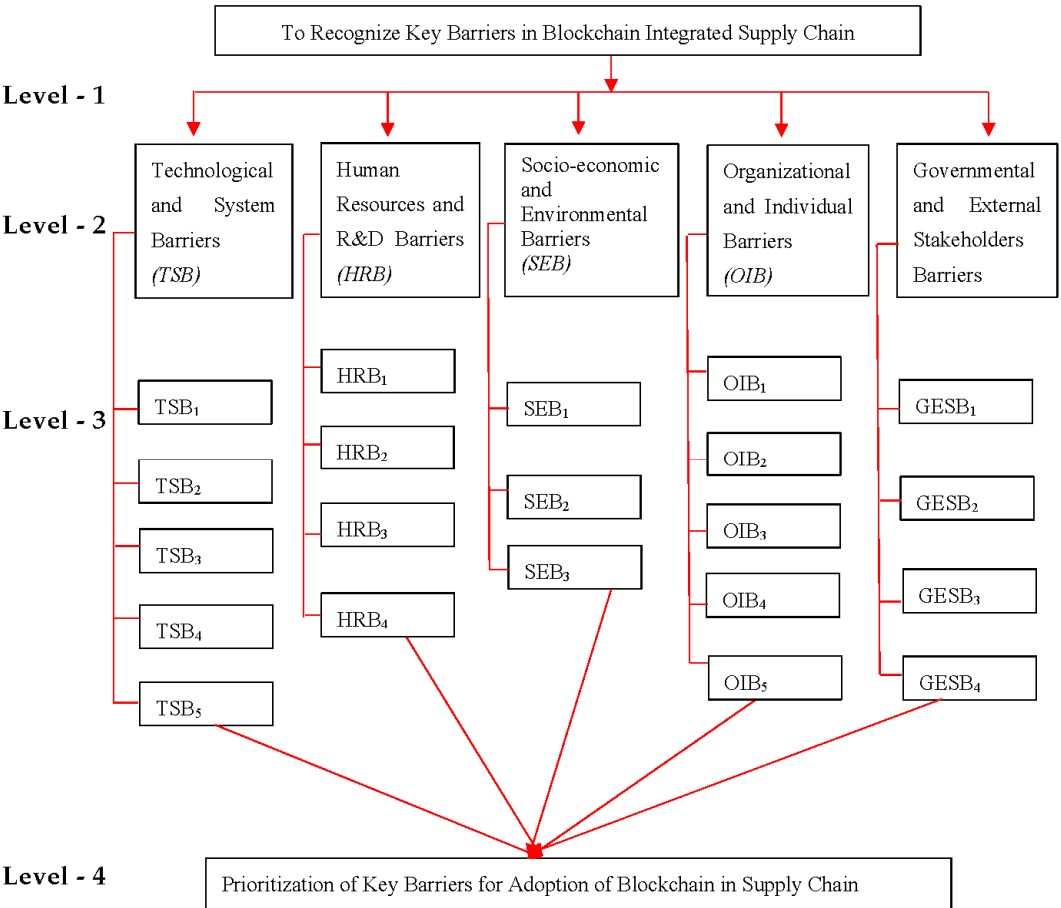

**Figure 4.** The hierarchical structure for prioritization of barriers in blockchain integrated supply chain.

### 4.1. Phase 1: Finalization of Barriers and Selection of Expert Team

Forty-one barriers relevant to the implementation of BCT in the textile sector were determined through an extensive review of the existing literature. The fuzzy Delphi approach was applied to address the uncertainty factors while finalizing the barriers. A team of experts were assigned to confirm the suitability and intensity level of identified barriers to adopt BCT in textile firms of Pakistan. In multi-criteria decision-making problems (MCDM), the selection of experts to gather information is a very complex task. The average number of experts to solve a (MCDM) problem is limited, and most of the studies used three, five, or seven experts for most of the (MCDM) techniques. Therefore, for our convenience we chosen ten experts using convenience sampling technique five in phase 1 and five in phase 2. The authenticity of the selected number of experts is in accordance with previous studies by different researchers in different sectors. The linguistic scale applied in this study is shown in Table 2.

The fuzzy Delphi approach was utilized to defuzzify the experts' input into crisp values. The results for finalizing barriers in blockchain integrated with supply chain are provided in Table 3. According to the existing literature and discussion with a relevant panel of experts, a threshold value was defined which adjusted from *r* > 0.60 to decide the addition or deletion of identified barriers. In addition, the experts were also requested to add or delete any barriers which they consider appropriate to the implementation of BCT in the textile industry. The changes were incorporated as per the experts' opinions in finalizing the process of the listed barriers. Based on the experts' inputs and results of the fuzzy Delphi method, 21 barriers were listed in this research study; these were further classified into five major barrier dimensions and were determined through focus group discussion and feedback. The identified dimensions are technological and system barriers (TSB), human resources and R&D barriers (HRB), socio-economic and environmental barriers (SEB), organizational and individual barriers (OIB) and governmental and external stakeholders' barriers (GESB). After identification of barrier categories, the weight vectors for the identified barriers were calculated in order to proceed to the next phase.

*4.2. Construction of Pairwise Comparison Matrix to Compute Weight of Criterion*

The weight vectors of key barriers were calculated using fuzzy AHP method. In this phase, the team of experts were consulted to establish a hierarchical framework of barriers as shown in Figure 2. The experts were then requested to provide input expressions to formulate a set of pairwise comparison matrices for the finalized key barriers and their sub-barrier categories using Saaty's 1–9 scale. The pairwise comparison matrix based on the judgment of an expert panel for the main barrier categories is recorded as follows:

$$E^1 = \begin{bmatrix} 1 & 3 & 1/5 & 3 & 7 \\ 1/3 & 1 & 1/7 & 5 & 3 \\ 5 & 7 & 1 & 3 & 5 \\ 1/3 & 1/5 & 1/3 & 1 & 1/7 \\ 1/7 & 1/3 & 1/5 & 7 & 1 \end{bmatrix}, E^2 = \begin{bmatrix} 1 & 9 & 3 & 1/7 & 1/5 \\ 1/9 & 1 & 1 & 3 & 3 \\ 1/3 & 1 & 1 & 1/5 & 3 \\ 7 & 1/3 & 5 & 1 & 5 \\ 5 & 1/3 & 1/3 & 1/5 & 1 \end{bmatrix},$$

$$E^3 = \begin{bmatrix} 1 & 9 & 3 & 3 & 1/3 \\ 1/9 & 1 & 5 & 1/5 & 3 \\ 1/3 & 1/5 & 1 & 7 & 5 \\ 1/3 & 5 & 1/7 & 1 & 3 \\ 3 & 1/3 & 1/5 & 1/3 & 1 \end{bmatrix}, E^4 = \begin{bmatrix} 1 & 3 & 1 & 1/3 & 1/5 \\ 1/3 & 1 & 3 & 3 & 7 \\ 1 & 1/3 & 1 & 5 & 1/3 \\ 3 & 1/3 & 1/5 & 1 & 1/5 \\ 5 & 1/7 & 3 & 5 & 1 \end{bmatrix}, E^5 = \begin{bmatrix} 1 & 3 & 3 & 5 & 1/7 \\ 1/3 & 1 & 5 & 7 & 3 \\ 1/3 & 1/5 & 1 & 3 & 1/7 \\ 1/5 & 1/7 & 1/3 & 1 & 3 \\ 7 & 1/3 & 7 & 1/3 & 1 \end{bmatrix}$$

The fuzzy pairwise comparison matrix of main barriers was formulated considering the inputs of experts by using Equation (5), which is provided in Table 4. The simple calculation procedure of fuzzy matrix when a criterion is compared with another criterion is illustrated, where symbol of *k depicts* the number of decision makers participated in the present research study.

For instance, $(\tilde{x}_{12}) = (a_{12}, b_{12}, c_{12}) = (3.00, 5.40, 9.00)$ as follows

$$a_{12=} \min_{k=1 \ to \ 5} (a_{ijk}) = min(a_{121}, a_{122}, a_{123}, a_{124}, a_{125},) = (3, 9, 9, 3, 3) = 3$$

$$b_{12} = \frac{1}{5} * \sum_{k=1}^{K} a_{ijk} = \frac{1}{5}(a_{121} + a_{122} + a_{123} + a_{124} + a_{125})$$
$$= \frac{1}{5}(3 + 9 + 9 + 3 + 3) = 5.40$$

$$c_{12} = \max_{k=1 \ to \ 5} \left(a_{ijk}\right) = max(a_{121}, a_{122}, a_{123}, a_{124}, a_{125}) = (3, 9, 9, 3, 3) = 9$$

After recording the inputs of experts, the next step is to compute the weights of barrier dimensions by applying the tool of fuzzy extent analysis. The pattern of computation for the weight vector of barriers is given below by using Equation (7) as follows:

$$H_{TSB} = (4.48, \ 12.308, \ 25.00) \otimes (1/115.66, 1/38.97, 1/14.10) = (0.039, 0.316, 1.773)$$

$$H_{HRB} = (4.45, 11.453, 20.33) \otimes (1/115.66, 1/38.97, 1/14.10) = (0.038, 0.294, \ 1.442)$$

$$H_{SEB} = (1.87, 8.178, 25.00) \otimes (1/115.66, 1/38.97, 1/14.10) = (0.016, 0.210, 1.773)$$

$$H_{OIB} = (1.62, 4.253, 23.00) \otimes (1/115.66, 1/38.97, 1/14.10) = (0.014, 0.109, 1.631)$$

$$H_{GESB} = (1.68, 2.780, 22.33) \otimes (1/115.66, 1/38.97, 1/14.10) = (0.015, 0.071, 1.584)$$

The above input expression of experts' assessments values are compared using Equation (8):

$$V(H_{TSB} \geq H_{HRB}) = 1.00, \ V(H_{TSB} \geq H_{SEB}) = 1.00, \ V(H_{TSB} \geq H_{OIB}) = 1.00, \ V(H_{TSB} \geq H_{GESB}) = 1.00$$

$$V(H_{HRB} \geq H_{TSB}) = 0.945, \ V(H_{HRB} \geq H_{SEB}) = 1.00, \ V(H_{HRB} \geq H_{OIB}) = 1.00, \ V(H_{HRB} \geq H_{GESB}) = 1.00$$

$$V(H_{SEB} \geq H_{TSB}) = 0.832, \ V(H_{SEB} \geq H_{SEB}) = 0.854, \ V(H_{TSB} \geq H_{OIB}) = 1.00, \ V(H_{TSB} \geq H_{GESB}) = 1.00$$

$$V(H_{TSB} \geq H_{HRB}) = 0.768, \ V(H_{TSB} \geq H_{SEB}) = 0.790, \ V(H_{TSB} \geq H_{OIB}) = 0.867, \ V(H_{TSB} \geq H_{GESB}) = 1.00$$

$$V(H_{TSB} \geq H_{HRB}) = 0.744, \ V(H_{TSB} \geq H_{SEB}) = 0.765, \ V(H_{TSB} \geq H_{OIB}) = 0.842, \ V(H_{TSB} \geq H_{GESB}) = 0.942$$

Next, the priority weight vectors of criteria are computed by using Equation (9) and Equation (10):

$$d'(A_1) = min(1.00, \ 1.00, 1.00, 1.00) = 1.00$$

$$d'(A_2) = min(0.945, \ 1.00, 1.00, 1.00) = 0.945$$

$$d'(A_3) = min(0.832, \ 0.854, 1.00, 1.00) = 0.832$$

$$d'(A_3) = min(0.768, \ 0.790, 0.867, 1.00) = 0.768$$

$$d'(A_3) = min(0.744, 0.765, 0.842, 0.942) = 0.744$$

Minimum weight

$$W' = (1.00, 0.945, 0.832, 0.768, 0.744)$$

Sum of Minimum weight

$$W' = (1.00 + 0.945 + 0.832 + 0.768 + 0.744) = 4.289$$

The final priority weights were computed using Equation (11) as follows:

$$W = \frac{1.00}{4.289}, \frac{0.945}{4.289}, \frac{0.832}{4.289}, \frac{0.768}{4.289}, \frac{0.744}{4.289} = (0.233, 0.220, 0.184, 0.179, 0.173\ )$$

$$W = (0.233, 0.220, 0.184, 0.179, 0.173\ )$$

Table 4 represents the intensity of the weight of the barrier dimensions and shows that 'technological and system barriers' obtained the top weight, followed by (TSB), human resources and R&D barriers (HRB), socio-economic and environmental barriers (SEB), organizational and individual barriers (OIB) and then governmental and external stakeholders' barriers (GESB).

The fuzzy pairwise matrix values were normalized by dividing each cell with the total (sum) of its column using Equation (1). The final criteria weights and ranking of barrier dimensions were obtained (See Table 5). The results suggest that the impact of technological and system-related barriers (TSB) have a higher weight amongst the five barrier dimensions. The weights for five barriers are 0.233, 0.220, 0.194, 0.179 and 0.173, respectively (See Table 5). The sum value of the weight vector should be equal to 1, which indicates consistency among the results. The consistency among the results can also be checked though consistency testing using fuzzy AHP method. If the aggregated matrix value of consistency is judged to be less than the threshold value of 0.10, then the criteria are highly consistent (see Table 6). In this study, the value is less than 0.10 of the main barriers; therefore, it is suggested that

no problem of inconsistency was found and we can proceed with further analysis of the barriers (See Figure 5).

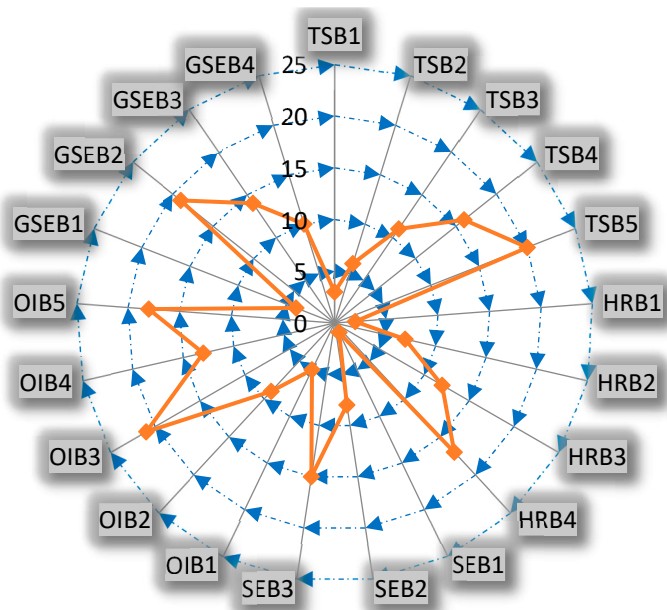

**Figure 5.** Ranking of barriers with normal weights.

*4.3. Sensitivity Analysis*

Sensitivity analysis is a tool which can be utilized to measure the uncertainty level in the final output of a mathematical model. This analysis is performed to test the stability of the priority ranking of alternative criterion in a multi-criteria decision-making model or framework. This analysis can be performed after the determination of optimal decision regarding a particular model under different case scenarios. A number of studies have used sensitivity analysis as an essential approach to confirm the authenticity of the model constructed [92,93]. In this research, slight changes in experts' expressions values can be treated as a target to determine the fluctuations among results using sensitivity analysis. The barrier 'technological and system (TSB)' is the most prioritized barrier, and human resources and R&D (HRB) was found to be the second priority barrier. This indicates that a slight change in the values of the weights of the finalized barriers may significantly influence the rest of the barriers. Therefore, technological and system (TSB) barriers weightage are fluctuated ranges from 0.233 (TSB) to (0.233 × 0.9 = 0.2097, 0.233 × 0.8 = 0.1864, 0.233 × 0.7 = 0.1631, 0.233 × 0.6 = 0.1398, 0.233 × 0.5 = 0.1165, 0.233 × 0.4 = 0.0932, 0.233 × 0.3 = 0.0699, 0.233 × 0.2 = 0.0466 and 0.233 × 0.1 = 0.0233, figures are recorded using four decimal points. Due to this modification, fluctuation can be seen in other barriers. The highest fluctuation recorded occurs in the 'human resources and R&D barriers' dimension (see Table 7). The final rating level of the key barriers also varies, respectively, as observed in Table 8. In addition, the sensitivity analysis can also be observed graphically as in Figure 6.

**Table 7.** Barrier values when changing technological and system barrier values.

| Barrier Categories | Normal Weight | | | | Incremental Changes | | | | |
|---|---|---|---|---|---|---|---|---|---|
| TSB | 0.233 | 0.210 | 0.187 | 0.163 | 0.140 | 0.117 | 0.093 | 0.070 | 0.047 | 0.023 |
| HRB | 0.220 | 0.225 | 0.231 | 0.236 | 0.242 | 0.247 | 0.253 | 0.259 | 0.265 | 0.271 |
| SEB | 0.194 | 0.199 | 0.203 | 0.208 | 0.213 | 0.218 | 0.223 | 0.228 | 0.233 | 0.239 |
| OIB | 0.179 | 0.183 | 0.188 | 0.192 | 0.196 | 0.201 | 0.206 | 0.210 | 0.215 | 0.220 |
| GSEB | 0.173 | 0.178 | 0.182 | 0.186 | 0.190 | 0.195 | 0.199 | 0.204 | 0.209 | 0.213 |

**Table 8.** The sensitivity analysis of sub-barriers with 'TSB' barrier changes from (0.233 × 0.9 ...... 0.233 × 0.1).

| | TSB = 0.233 Normal | TSB = 0.210 | TSB = 0.187 | TSB = 0.163 | TSB = 0.140 | TSB = 0.117 | TSB = 0.093 | TSB = 0.070 | TSB = 0.047 | TSB = 0.023 |
|---|---|---|---|---|---|---|---|---|---|---|
| $TSB_1$ | 3 | 4 | 5 | 5 | 5 | 7 | 8 | 10 | 13 | 17 |
| $TSB_2$ | 6 | 6 | 7 | 9 | 10 | 10 | 11 | 14 | 16 | 18 |
| $TSB_3$ | 11 | 11 | 13 | 13 | 15 | 16 | 16 | 19 | 19 | 19 |
| $TSB_4$ | 16 | 17 | 18 | 18 | 19 | 20 | 19 | 20 | 20 | 20 |
| $TSB_5$ | 20 | 21 | 21 | 21 | 21 | 21 | 21 | 21 | 21 | 21 |
| $HRB_1$ | 2 | 2 | 2 | 2 | 2 | 2 | 2 | 2 | 2 | 2 |
| $HRB_2$ | 7 | 7 | 6 | 6 | 6 | 5 | 5 | 5 | 5 | 5 |
| $HRB_3$ | 12 | 12 | 11 | 11 | 11 | 11 | 9 | 9 | 9 | 9 |
| $HRB_4$ | 17 | 16 | 16 | 16 | 16 | 15 | 14 | 15 | 14 | 13 |
| $SEB_1$ | 1 | 1 | 1 | 1 | 1 | 1 | 1 | 1 | 1 | 1 |
| $SEB_2$ | 8 | 8 | 8 | 7 | 7 | 6 | 6 | 6 | 6 | 6 |
| $SEB_3$ | 15 | 15 | 15 | 15 | 14 | 14 | 12 | 13 | 12 | 12 |
| $OIB_1$ | 5 | 5 | 4 | 4 | 4 | 4 | 4 | 4 | 4 | 4 |
| $OIB_2$ | 9 | 9 | 9 | 8 | 8 | 8 | 7 | 7 | 7 | 7 |
| $OIB_3$ | 21 | 20 | 20 | 20 | 20 | 19 | 18 | 18 | 18 | 16 |
| $OIB_4$ | 13 | 13 | 12 | 12 | 12 | 12 | 10 | 11 | 10 | 10 |
| $OIB_5$ | 18 | 18 | 17 | 17 | 17 | 17 | 15 | 16 | 15 | 14 |
| $GSEB_1$ | 4 | 3 | 3 | 3 | 3 | 3 | 3 | 3 | 3 | 3 |
| $GSEB_2$ | 19 | 19 | 19 | 19 | 18 | 18 | 20 | 17 | 17 | 15 |
| $GSEB_3$ | 14 | 14 | 14 | 14 | 13 | 13 | 17 | 12 | 11 | 11 |
| $GSEB_4$ | 10 | 10 | 10 | 10 | 9 | 9 | 13 | 8 | 8 | 8 |

A similar calculation can be performed to check the fluctuation among the second highest ranked barrier named 'human resources and R&D barriers (HRB)'; the findings are shown in Table 9. The graphical representation of the second priority barrier is sketched in Figure 7. The human resources and R&D barrier weights are slightly modified from 0.220 (HRB) to (0.220 × 0.9 = 0.1980, 0.220 × 0.8 = 0.1760, 0.220 × 0.7 = 0.1540, 0.220 × 0.6 = 0.1320, 0.220 × 0.5 = 0.1100, 0.220 × 0.4 = 0.0880, 0.220 × 0.3 = 0.0660, 0.220 × 0.2 = 0.0440 and 0.220 × 0.1 = 0.0220, values. These findings indicate that the highest fluctuation occurs in the 'technological and system barrier (TSB)' dimension (see Table 10).

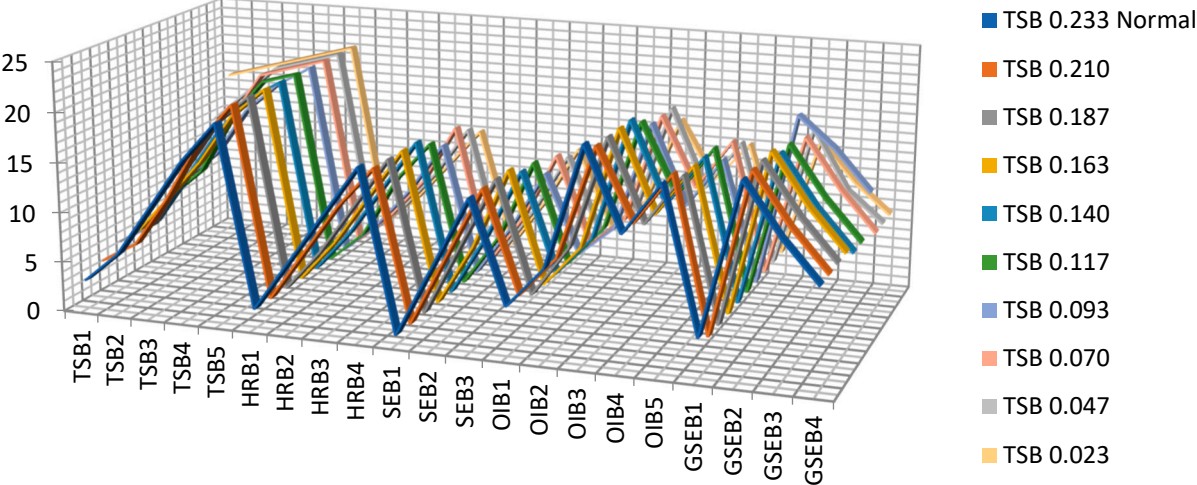

**Figure 6.** The results of sensitivity analysis for technological and system barriers (TSB).

**Table 9.** Barrier values when changing human resources and R&D barrier values.

| Barrier Categories | Normal Weight | | | | Incremental Changes | | | | | |
|---|---|---|---|---|---|---|---|---|---|---|
| TSB | 0.233 | 0.238 | 0.244 | 0.249 | 0.254 | 0.260 | 0.266 | 0.272 | 0.278 | 0.284 |
| HRB | 0.220 | 0.198 | 0.176 | 0.154 | 0.132 | 0.110 | 0.088 | 0.066 | 0.044 | 0.022 |
| SEB | 0.194 | 0.198 | 0.203 | 0.207 | 0.212 | 0.216 | 0.221 | 0.226 | 0.231 | 0.236 |
| OIB | 0.179 | 0.183 | 0.187 | 0.191 | 0.195 | 0.200 | 0.204 | 0.209 | 0.213 | 0.218 |
| GSEB | 0.173 | 0.177 | 0.181 | 0.185 | 0.189 | 0.193 | 0.198 | 0.202 | 0.207 | 0.211 |

**Table 10.** Sensitivity analysis of sub-barriers with 'HRB' barrier changes from (0.220 × 0.9 ... ... 0.220 × 0.1).

| | HRB = 0.220 Normal | HRB = 0.198 | HRB = 0.176 | HRB = 0.154 | HRB = 0.132 | HRB = 0.110 | HRB = 0.088 | HRB = 0.066 | HRB = 0.044 | HRB = 0.022 |
|---|---|---|---|---|---|---|---|---|---|---|
| $TSB_1$ | 3 | 3 | 2 | 2 | 2 | 2 | 2 | 2 | 2 | 2 |
| $TSB_2$ | 6 | 6 | 6 | 6 | 5 | 5 | 5 | 5 | 5 | 5 |
| $TSB_3$ | 11 | 11 | 11 | 11 | 10 | 10 | 10 | 9 | 9 | 9 |
| $TSB_4$ | 16 | 16 | 16 | 16 | 15 | 15 | 15 | 14 | 14 | 14 |
| $TSB_5$ | 20 | 20 | 20 | 20 | 19 | 19 | 18 | 18 | 17 | 17 |
| $HRB_1$ | 2 | 2 | 4 | 5 | 6 | 6 | 8 | 10 | 12 | 12 |
| $HRB_2$ | 7 | 8 | 9 | 10 | 11 | 12 | 14 | 16 | 18 | 19 |
| $HRB_3$ | 12 | 13 | 13 | 15 | 16 | 17 | 19 | 20 | 20 | 20 |
| $HRB_4$ | 17 | 18 | 18 | 19 | 21 | 21 | 21 | 21 | 21 | 21 |
| $SEB_1$ | 1 | 1 | 1 | 1 | 1 | 1 | 1 | 1 | 1 | 1 |
| $SEB_2$ | 8 | 7 | 7 | 7 | 7 | 7 | 6 | 6 | 6 | 6 |
| $SEB_3$ | 15 | 15 | 15 | 14 | 14 | 14 | 13 | 13 | 13 | 13 |
| $OIB_1$ | 5 | 5 | 5 | 4 | 4 | 4 | 4 | 4 | 4 | 4 |
| $OIB_2$ | 9 | 9 | 8 | 8 | 8 | 8 | 7 | 7 | 7 | 7 |
| $OIB_3$ | 21 | 21 | 21 | 21 | 20 | 20 | 20 | 19 | 19 | 18 |
| $OIB_4$ | 13 | 12 | 12 | 12 | 12 | 11 | 11 | 11 | 10 | 10 |
| $OIB_5$ | 18 | 17 | 17 | 17 | 17 | 16 | 16 | 15 | 15 | 15 |
| $GSEB_1$ | 4 | 4 | 3 | 3 | 3 | 3 | 3 | 3 | 3 | 3 |
| $GSEB_2$ | 19 | 19 | 19 | 18 | 18 | 18 | 17 | 17 | 16 | 16 |
| $GSEB_3$ | 14 | 14 | 14 | 13 | 13 | 13 | 12 | 12 | 11 | 11 |
| $GSEB_4$ | 10 | 10 | 10 | 9 | 9 | 9 | 9 | 8 | 8 | 8 |

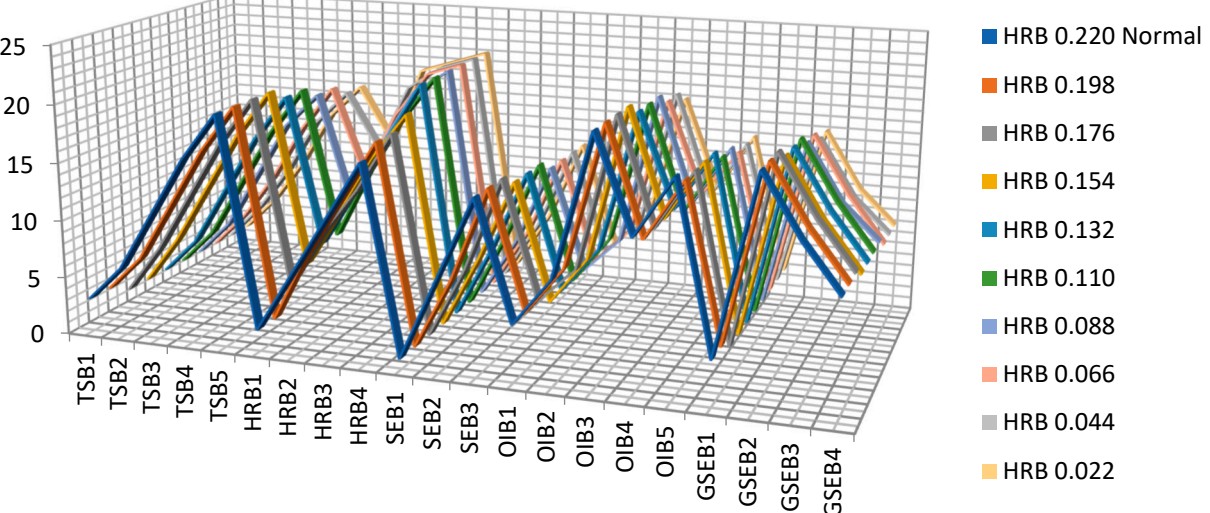

**Figure 7.** Results of sensitivity analysis for human resources and R & D barriers (HRB).

## 5. Discussion of Findings

This section elaborates the key findings and results discussion to assist concerned stakeholders in making strategic decisions for adopting blockchain technology within the textile supply chain. The following sections provide a description of the results concerning prioritized barriers.

### 5.1. Key Barrier Dimensions

The findings of this research study indicate that the 'technological and system barrier' was found to be the first priority among the barrier dimensions as shown in Table 6. The technological and system barrier dimension is an essential barrier for BCT execution in order to achieve sustainable supply chain excellence. The human resources and R&D barrier dimension was found to be second in priority value. The socio-economic and environmental barrier dimension was ranked third. The organizational and individual barrier dimension obtained fourth rank and government and external stakeholder barrier dimension was ranked fifth in the barrier dimensions. The highly prioritized values used to consider the most important BCT-related barriers were ranked such that TSB > HRB > SEB > OIB > GESB which is shown in Table 5. Finally, it can be observed that technological and system barriers (TSB) were considered high priorities by firms executing BCT.

### 5.2. Barriers Ranking for BCTSC Adoption in Pakistani Textile Industries

The overall barrier rankings for BCTSC adoption using fuzzy AHP method is shown in Table 6. The analysis results of all finalized barriers and sub-barrier dimensions are individually discussed in the below sub-sections:

### 5.2.1. Technological and System Barriers (TSB)

In blockchain technology adoption, there are a certain number of barriers, i.e., the lack of technical capabilities regarding security, systems, capacity, capabilities and personnel competencies. These are usually taken as essential aspects of achieving business excellence. The technological and system barriers were found to be the highest rank as compared to other peer-to-peer barriers. This illustrates that textile firms operating their businesses in Pakistan are unable to execute sophisticated technological systems due to lack of popularity of BCT implementation. In our research, this dimension categorized five sub-barriers which indicates that technological and system barriers sub-criteria were $TSB_1 > TSB_2 > TSB_3 > TSB_4 > TSB_5$. This shows that immaturity of technology was found to be the highest-ranked barrier, and the risk of cyber-attacks and lack of expertise and technical support in IT was the less-preferred barrier of this category. The results of this dimension are in line with the findings of some previous studies. Therefore, the stakeholders who assume that blockchain adoption is complicated and difficult to understand may show more resistance towards innovation and system acceptance.

### 5.2.2. Human Resources and R&D Barriers (HRB)

In this dimension of barriers, the rating of the human resources and R&D barriers was $HRB_1 > HRB_2 > HRB_3 > HRB_4$ (see Table 6), respectively, in which negative perception of IT among workers' minds considered the higher weight value of a barrier and lack of research and development departments' initiative on adoption of BCT was the lowest for this section. The previous literature has consistently described that technologically immature, incompatible and the negative mindsets of employees are the major hurdles in adopting BCT at a larger scale. From these findings, we conclude that higher levels of motivation in terms of monetary and non-monetary incentives can change the mindset of workers and lead them towards technological acceptance.

### 5.2.3. Socio-Economic and Environmental Barriers (SEB)

Socio-economic and environmental barriers' priority vector ranked as $SEB_1 > SEB_2 > SEB_3$ (Table 6), respectively, in which poor economic behavior in the long run was found

to be the highly prioritized barrier, and neglecting ecological standards was the lowest one for this section. Due to a scarcity of qualified employees, there is a need to recruit human resources and acquire skills through outsourcing for successful adoption of BCT. The financial cost is not only limited to the installation of technology equipment's but also to the retainment of competent staff. The pre-existing literature on socio-economic and environmental factors indicates that higher costs, poor environmental standards and skilled labor can increase chances for resistance from stakeholders to adopt BCT.

### 5.2.4. Organizational and Individual Barriers (OIB)

The ranking values for organizational and individual barriers were $OIB_1 > OIB_2 > OIB_4 > OIB_5 > OIB_3$ (see Table 6), respectively, in which a lack of commitment from management and support for blockchain integration was highly prioritized among the barriers, and lack of new organizational policies for adopting technology was the lowest one for this section. In comparison to other network technologies, such as Visa or PayPal, blockchain is still underdeveloped in many countries like Pakistan, Bangladesh, Sri Lanka, Turkey, Egypt, Nepal and India.

### 5.2.5. Governmental and External Stakeholder Barriers (GESB)

Ranking values for governmental and external stakeholder barriers were $GSEB_1 > GSEB_4 > GSEB_3 > GSEB_2$ (see Table 6), respectively, in which unclear issues of taxation and regulatory uncertainties for blockchain integration were found as the highly prioritized barriers and lack of government interest in blockchain was the lowest for this section. For developing countries, the acceptance of technology is a difficult task; therefore, the management of textile organizations must be vigilant concerning the negative mindsets of employees and potential resistance from the workforce.

### 5.3. Comparison with the Existing Literature

It is very important to compare the barriers listed in this present study with pre-existing studies in order to adopt BCT in the SSC of the textile sector [94,95]. The prioritization of barriers may fluctuate from one sector to another sector due to socio-economic and cultural constraints. In this study, an integrated framework based on the fuzzy Delphi and AHP techniques was developed to rate the highly significant barriers related to BCT adoption. The technological and system related barrier (TSB) has been ranked as a highly prioritized barrier for blockchain implementation. The small textile industry of Pakistan is facing a lack of expertise and technical support in IT. Secondly, the human resources and R&D related barrier (HRB) ranks as a highly prioritized factor as there exists a negative perception of IT among workers' minds. Much of the industry lacks professional technical labor and R&D departments. The socio-economic and environmental barrier (SEB) dimension was ranked third in terms of priority in this research study. Öztürk and Yildizbaşi, [96] discussed these barriers in the context of four different sectors, i.e., construction, logistics, agriculture and health. The results reveal that decision makers should focus on the implementation of blockchain-related practices and explore potential barriers that may influence the entire textile supply chain in achieving sustainable goals. Therefore, the textile industry in Pakistan is in dire need of developing blockchain-based strategies that could help concerned stakeholders in tackling potential barriers, which were scientifically evaluated, prioritized and discussed, through this research. This study bridges the research gap by (1) identifying the key barriers in implementation of BCTSC using fuzzy Delphi-AHP technique, (2) developing a skeletal model of BCTSC activities that may help policy makers and concerned stakeholders to adopt BCT activities in their business domains and (3) providing research implications from the local context to the global context in a more systematic way.

## 6. Implications of the Research

The insights from this research would be helpful for stakeholders and top industrial management in formulating policies regarding implementation of BCT-related practices

in the existing supply chains. In this regard, the following important implications may be made, which are summarized below:

*6.1. Managerial Implications*

- Implementing BCT into Supply Chain and Arranging a Training Program

The technological and system barrier (TSB) is an essential barrier for BCT execution in order to achieve sustainable supply chain excellence. It is important for textile brands to prioritize the integration of blockchain technology in the supply chain. Developing countries are in a transition phase with regards to adopting blockchain technology; therefore, they need to establish R & D departments. Training and apprenticeship programs on BCT-based practices may help industrial managers to strengthen the supply chain system in order to achieve sustainable goals.

- Empowering Human Resource Departments and Establishing R&D Units

The human resources and R&D barrier (HRB) dimension was found to be second in priority value after the technological and system barrier dimension. It is very important to set up R&D departments and strategically align the working of these departments with the goals of the organizations by empowering the human resource department. Human resource departments need to develop policies in such a way that the negative perception of IT among workers' minds can be discouraged and a spirit of research and development can be encouraged.

- Formulating Strategic Policy for Implementing BCT-Based Practices in the Supply Chain

The socio-economic and environmental barrier (SEB) dimension was ranked third after the human resources and R&D barrier dimension. The findings of this research would be helpful for concerned companies to formulate a strategic policy based on socio-economic objectives. Therefore, it is crucial to execute BCT-based practices and more innovative methods in the textile industry to ensure data safety and security measures.

- Understanding the Science of Peoples and Organizational Dynamics

The organizational and individual barrier (OIB) dimension obtained fourth rank. It is necessary to understand employee behavior towards technology adoption and check to what extent environmental standards are implemented. In addition, commitment from top management and support for blockchain integration with the supply chain is crucial for improving the supply chain performance.

- Convincing Government and External Stakeholders Regarding the Benefits of BCT

The government and external stakeholder barrier (GESB) dimension was ranked fifth in the barrier dimensions. Government and external supply chain partners need to be sensitized regarding the advantageous features/characteristics of blockchain technology. For developing countries, the acceptance of technology is a difficult task; therefore, negative mindsets and resistance from the workforce should be managed very cautiously. The labor force is the main resource of any organization which can resist change; thus, personnel of these organizations need to be well-trained so they can prove themselves to be valuable assets for the organization.

*6.2. Social Implications*

The textile firms operating their businesses in different areas are facing socio-economic and geographical constraints. In light of the data gathered through field visits and consultation with experts, it has been suggested that the top priority should be to change the mindset of non-managerial staff, operational management and middle management in technology adoption. The decision makers and industrialists must concentrate on procedures that encourage the labor force towards adoption of blockchain practices in supply chain management and ensure them that this will not harm them. Therefore, the firms must arrange regular training sessions regarding the significance of blockchain practices with

the concerned management in order to minimize the chances of conflicts and resistance due to negative mindsets. In this context, the labor unions and management of the firms should have consensus regarding barrier eradication.

### 6.3. Global Implications

In today's competitive global marketplace, the adoption of BCT has been considered key for achieving sustainable business goals. The contemporary textile supply chain systems are getting larger and more complicated due to the involvement of global partners from the entire chain from upstream towards downstream. Presently, stakeholders are concerned about the adoption of BCT-related practices in the textile supply chain. In terms of research, innovation and technological advancements, Pakistan is trailing far behind Turkey, China, the Philippines, Vietnam, Egypt, and India. South Asian countries have great potential and have achieved success in the domain of information communication technologies. Due to its popularity, the Ministry of Information Technology and Telecommunication (MOITT) in Pakistan emphasized the need to establish a special economic zone (SEZ) to support the manufacturing sector, especially the textile sector, by introducing technology SEZs. M-3 Industrial City is the largest SEZ of Pakistan and was launched as a landmark project to support the development of the manufacturing sector with an excellent environment for competing in the global market. MOITT designed the Digital Pakistan Policy, which was further sanctioned by parliament in 2018. This policy states the guidelines that will strengthen the sophisticated IT-based ecosystem and improve the overall growth in the country, whereas previously, garment manufacturers in Pakistan were not conscious of BCT due to a general lack of awareness of BCT, a lack of technological advancement, and a threat of financial loss due to the implementation of BCT. Currently, textile enterprises are becoming aware of BCT-related practices because it facilitates capturing the share of the global market, providing transparent information to customers and ultimately enhancing sales value through better brand positioning. Due to much pressure from global stakeholders, garment manufacturers are paying attention and considering the incorporation of BCT activities along the supply chain. Currently, there is no single system available across the globe to keep records of a product's history throughout the holistic supply chain. Therefore, traceability of product lines and data size among supply chain partners is quite difficult due to the present number of stages. This study also has significant global implications for BCT adoption in the textile industries of different countries. Researchers can utilize this proposed model in their countries, rather than reinventing the wheel, as this study focused specifically on the context of Asia. A similar type of research framework can be developed, tested and applied to integrate BCT with sustainable supply chains of developed countries as well. Undeniably, the advantages are obvious, but there are still a series of challenges that prevail in textile manufacturing organizations. The vertically-oriented textile supply chain organizations are encouraged to develop the first-move strategy of BCT implementation in textile supply chains, which may help them obtain a dominant position in the global marketplace. In our study, the proposed framework may also be considered as a viable source of strategy development in transition to Industry 4.0. The real-time adoption of BCT into sustainable supply chain practices can open a new door to empirical research for reshaping business models.

## 7. Conclusions

The basic purpose of this research is to evaluate the blockchain technology-related barriers and develop an integrated model for sustaining business continuity for a longer time period in the textile sector. This research study includes five major components: (i) the determination of barriers to execute the blockchain technology in the textile supply chain through fuzzy Delphi, (ii) calculation of weight vector using fuzzy AHP, (iii), analysis of barriers along with normal weights (iv) fluctuation of the barrier ranking using sensitivity analysis, and (v) managerial insights for the concerned stakeholders. Practically, the experts in the relevant fields have encountered many challenges in making accurate decisions for

real-life problems due a to lack of appropriate solutions and opportunities. Therefore, this research developed the hybrid model of fuzzy Delphi method based on fuzzy AHP to rank barriers. Fuzzy Delphi was applied to select or reject the criterion through threshold value 0.60, whereas fuzzy AHP technique was applied to obtain the weight vector for finalized barriers. The sensitivity analysis was used to confirm the authenticity of the model and variations among criterion weight values. The findings and results indicate that top level management should pay more attention and encourage bottom level management in adopting the technological and system- related practices. In addition, technological transformation is very complex for the textile and clothing sectors due to a certain number of constraints. The execution of BCT would be the source of sustainable strategic development from transition to Industry 4.0 generation. In this study, the rank order of identified barriers is sorted as TSB > HRB > SEB > OIB > GESB. Similarly, the weight vectors and ranking of sub-barriers can also be computed. It can be observed that technological and system barriers (TSB) have attained a significant level of importance. The socio-economic and environmental barrier (SEB) dimension was ranked third l after human resources and R&D barrier (HRB) dimension. The organizational and individual barrier (OIB) dimension obtained fourth rank and government and external stakeholder barrier (GESB) dimension was ranked fifth in the barrier dimensions. The findings of the research results confirm that the technological oriented supply chain performs the major roles among all stakeholders for dissemination of information in the holistic chain. Therefore, concerned stakeholders need to pay attention to the identification of barriers, spend more money and invest in IT-based resources.

Despite the significance of this research, the shortcomings of this study are the evaluation phase of recognizing the barriers which was very challenging. Also, this study was limited to the specific geography of Pakistan in the garment division of the textile sector. Although this study was targeted to identify key barriers, if implemented by stakeholders, there may exist some new and obsolete barriers due to the passing of time, technological advancements, and structural reforms. For future research direction, this study can be extended using an empirical based approach to make comparisons with other developing countries using econometric models, structural equation modeling and statistical tools. Researchers can utilize this proposed model in their countries rather than reinventing the wheel as this study specifically focused on the context of Asia. Furthermore, the ranked barriers may also be analyzed to compare with fuzzy set qualitative comparative analysis and construct their inter-relationships, using ISM and DEMATEL methods etc. which is a novel method.

**Author Contributions:** Conceptualization, M.N., M.H., M.A.Z., F.M.N. and M.U.; methodology, M.N.; validation of data, M.N., L.Y., M.H. and M.U.; writing-review and editing, F.M.N. and L.D.; proof reading M.Y.M. and L.Y. All authors have read and agreed to the published version of the manuscript.

**Funding:** This research received no external funding.

**Institutional Review Board Statement:** Not applicable.

**Informed Consent Statement:** Not applicable.

**Data Availability Statement:** Please contact the authors via email for the data.

**Acknowledgments:** The authors would like to thank the Editor and the anonymous referees for their constructive comments to improve the quality of this research.

**Conflicts of Interest:** The authors declare no conflict of interest.

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
