# Peer review of "Devising a Mechanism for Analyzing the Barriers of Blockchain Adoption in the Textile Supply Chain: A Sustainable Business Perspective"

_sustainability, doi:10.3390/su142316159_

Round 1

Reviewer 1 Report

Manuscript Number: Sustainability-1976518

Title of the paper: Devising a Mechanism for Traceability and Sustainability in a Supply Chain: A Blockchain Diffusion Perspective

Reviewers Comments: Authors analyzed the key barriers in adopting Blockchain related practices within the textile organization. The field is interesting. However, the manuscript needs comprehensive corrections. The reviewer suggests a major correction according to the following remarks.

Comment 1: The current abstract is descriptive one. It should be written in quantitative way.

Comment 2: The introduction needs elaboration. It should set out the context, summarize recent research related to the topic, highlights gap in current literature, and should thoroughly introduce the work of the paper.

Comment 3: The aim and novelty of this study should be mentioned clearly in the end of the introduction.

Comment 4: The contribution of the application with to the literature should be further emphasized.

Comment 5: The advantage of using the proposed methodology (Fuzzy Delphic-Fuzzy AHP) in real world problems should further be exemplified.

Comment 6: A comparison with existing techniques on how they behave for real world problems could be made to validate the research.

Comment 7: I highly recommend a comprehensive readout and revision to improve the quality and language of the manuscript.

Comment 8: Would you explicitly specify the novelty of your review? What progress against the most recent research on Blockchain diffusion perspective in supply chain for sustainability?

Comment 9: All the tables and figures should be further improved.

Comment 10: The numerical results should be discussed in more detailed and critical way.

Comment 11: Results and Discussion is not properly discussed. Much more explanations and interpretations must be added for the results and discussion.

Comment 12: Please present conclusion more in-depth and elaborate on the significance of the findings. In order to make the conclusion section more clearly, authors are highly encouraged to include the point-by-point findings of this article.

Comment 13: Please talk about the limitations and future work more in detail in the conclusion section.

Comment 14: Grammar of the manuscript should be checked.

Comment 15: Check the reference section.

Comment 16: More references from Sustainability Journals (MDPI) can be considered.

Reviewer 2 Report

Summary

If I understand the authors correctly, the article conducts a literature review, with the purpose of identifying barriers to the adoption of blockchain technology in the textile industry. Then, the authors confirm the findings from the literature in the textile industry in Pakistan, based on a Delphi study.

SWOT

Although it is practically important and relevant to identify barriers to the adoption of blockchain technology in the textile industry, the article in its current state is really difficult to read, often repetitive, lacks a clear storyline, and does not clearly distinguish between main issues and side issues. It is not clear from the introduction what the precise aim of the paper is (what is the scope of the study, who is the intended audience, what exactly is the article contributing, and to which debate (what are the gaps addressed?), what the precise approach is (e.g., how data were collected, from whom, how, what kind of data?), and why this approach is the most appropriate approach). It is not transparent in terms of how it reviewed the literature (which literature was selected) and how the Delphi study was conducted. As a result, it remains entirely unclear how the recommendations and conclusions relate to the study, and it cannot be established whether these results are valid, or to what extent they can or cannot be generalized.

Major issues

To address the issues mentioned above, I suggest better structuring, and cleaning up the article. Try to be very clear about the scope and intended contributions of the study (are its findings only valid for the textile industry (or manufacturing?) in Pakistan or developing countries, and if so, why is this so? What limits its generalization? Is this intended, or a result of the methodology?). The research questions must correspond to this chosen scope, which currently does not appear to be the case.

The theoretical background currently consists of discussions of a range of loosely related topics, and a real storyline is missing. Part of it seems to be focusing on sustainability issues, but it is not entirely clear how this relates to the research objectives and questions. Please explain clearly what the purpose of this chapter is, and how it is achieved. This could be done by using some binding and meta text, that guide the reader through the article.

The literature review is unstructured: if the purpose of the article is to review literature about barriers to blockchain adoption, the methodology of the review needs to be explicated.

The methodology section appears more like a chapter from a textbook than a description of the methodology used by the authors. Please clearly explain what the purpose of the study is, and demonstrate how the chosen method helps to achieve this purpose.

To what extent is the selected case representative of the industry, and are the findings valid? It is currently not possible to verify this. What exactly is your sample, who is included, based on which inclusion or exclusion criteria? How many experts participated, who are they? Etc. What are the identified barriers? How are they clustered? Who did this and based on which information?

To me it is entirely unclear how the recommendations relate to the study’s findings. The explanation is not transparent.

Minor Issues

Title

The title is rather confusing and does not seem to correspond to what is actually done in the article. Please revise.

Many Figures and tables are very difficult to read because of size issues (they are larger than the space available in the document). Please review and adjust.

Abbreviations

In the document abbreviations are used that are not introduced: please make sure all abbreviations are explained or written in full the first time.

Language

The language of the article is very woolly and often very difficult to understand, as a result of errors in word use, syntax and grammar. I suggest having a knowledgeable native speaker proofread and edit the article before publication.

Font issues

The font and font size seem to change in several places in the text, sometimes inside a citation. Please correct.

Numbering

The numbering of the paragraphs and sub paragraphs is inconsistent. Please check.

Reviewer 3 Report

This paper highlights the need for an inclusive understanding of the various technological, environmental and socio-economic perspectives to create blockchain applications that work for textile sectors. The comments are as follows:

1. Use the full form of all the abbreviations in their first use. for example it is written BCT in the abstract. what is BCT?

2. In the abstract, Write about the result and managerial implications of the study.

3. Introduction should be written in precise form.  

4. All the Figures are here and there not as a group and also not able to understand. kindly make a group.

6. Recently Many papers have been published in the area of blockchain and sustainable supply chains such as DOI: 10.4018/IJISSCM.2018070104 which need to be updated in the Introduction and literature review section. 

5.  Why authors have used data in the fuzzy form? and what will happen if data will use in crisp or grey form.- data need to furnish.

6.  Why the authors have used tringular fuzzy? What are the different types of fuzzy? Data needs to be furnished.

6.  

5.  

Round 2

Reviewer 1 Report

I am satisfied with the responses provided by the authors in the revised manuscript. In the current form, it may be considered for publication in Sustainability.

Reviewer 2 Report

Review Sustainability 1976518 R1

I have read the revision with interest. It has made clearer what the level of analysis is, and from which perspective the article has been written: It identifies barriers to the adoption of blockchain technology, from the perspective of the textile organization (in an emerging economy), based on a review of the literature. My strong recommendation here is to maintain and increase consistency throughout the text. Explain better what is done, how it is done, and why.

Major Issues

The article remains in dire need of language editing. There are major problems in terms of sentence structure, correspondence between subject and verb, grammar, vocabulary, etc. I strongly advise a knowledgeable native speaker or (preferably) a professional text-editor, who understands the topic, edits the text.  This should lead to a major rewrite of the article, as there are innumerable issues that need to be corrected and thus changed.

The title is now more focused on what the article contributes. Apart from the incorrect use of the word 'diffusion', it is now a clear and appropriate title.

Methods

Although the results have some face validity, the article remains insufficiently transparent in terms of its methodology. How the authors reviewed the literature (Which literature was selected? How (search string? including 'Pakistan'?)? Which inclusion/exclusion criteria were used? How was the literature analyzed (content analysis?), i.e., where do the categories and the barriers in Table 1 come from? The fuzzy Delphi approach, or the literature) and how the Delphi study was conducted: Which criteria were used to select the experts? What were they asked to do, and how was this data then analyzed? Please explain the steps used in the fuzzy AHP as visualized in Figures 2 and 3. What is the precise approach (e.g., how data were collected, from whom, how, what kind of data?), and why this approach is the most appropriate approach). To me it remains unclear how the recommendations and conclusions relate to the study, and it cannot be established whether these results are valid, or to what extent they can or cannot be generalized.

To what extent is the selected case representative of the industry, and are the findings valid? It is currently not possible to verify this. What exactly is your sample, who is included, based on which inclusion or exclusion criteria? How many experts participated, who are they? Etc. What are the identified barriers? How are they clustered? Who did this and based on which information?

Minor issues:

In the title, and in the abstract (first line), but also in the research questions and elsewhere, the authors incorrectly use 'diffusion', where I would expect 'adoption'. This is rather confusing. Please have the text edited and corrected.

Reviewer 3 Report

The authors have incorporated almost all the comments. the manuscript may be accepted in its current form.

Round 3

Reviewer 2 Report

Review v3

The authors have made a substantial effort to improve the article. The structure and methodology followed have become much clearer. Many of the issues identified in the previous reviews have been corrected. Nonetheless some issues remain (of which the language issue is currently the most important and serious one). In no particular order, they are:

References

References are not in APA7 and several are incomplete. Please revise.

Title

The new title much better represents the actual contents of the article, but I suggest to take out ‘Devising a mechanism for’ from the title (as this is not what the authors do). Please revise.

Readability

Some formulas on p. 15 (l. 632-635) are very difficult to read because of font size issues. Please review and adjust.

Language

The language of the article remains very woolly and often very difficult to understand, as a result of many errors in word use, syntax and grammar. I suggest having a knowledgeable native speaker proofread and edit the article before publication.

Scope

I think it is still possible to further improve upon the clarity of the scope and intended contributions of the study. Are its findings only valid for the textile industry (or manufacturing?) in Pakistan or developing countries, and if so, why is this so? I suggest to clearly and unambiguously express this in the abstract and the introduction.